# Directional integration and pathway enrichment analysis for multi-omics data

Mykhaylo Slobodyanyuk [1,2,6], Alexander T. Bahcheli [1,3,6], Zoe P. Klein [1,3], Masroor Bayati [1,2], Lisa J. Strug [4,5] & Jüri Reimand [1,2,3] ✉

Omics techniques generate comprehensive profiles of biomolecules in cells and tissues. However, a holistic understanding of underlying systems requires joint analyses of multiple data modalities. We present DPM, a data fusion method for integrating omics datasets using directionality and significance estimates of genes, transcripts, or proteins. DPM allows users to define how the input datasets are expected to interact directionally given the experimental design or biological relationships between the datasets. DPM prioritises genes and pathways that change consistently across the datasets and penalises those with inconsistent directionality. To demonstrate our approach, we characterise gene and pathway regulation in *IDH*-mutant gliomas by jointly analysing transcriptomic, proteomic, and DNA methylation datasets. Directional integration of survival information in ovarian cancer reveals candidate biomarkers with consistent prognostic signals in transcript and protein expression. DPM is a general and adaptable framework for gene prioritisation and pathway analysis in multi-omics datasets.

High-throughput omics technologies enable systematic mapping of genes, transcripts, proteins, and epigenetic states in cells. While data generation methods advance rapidly, data interpretation remains challenging as our understanding of complex molecular pathways and interaction networks is limited. Pathway enrichment analysis[1] is a common technique to interpret omics datasets using existing knowledge of gene function and biological processes. It examines candidate gene lists detected in omics experiments to identify significantly enriched biological processes or molecular pathways to explain the underlying experimental conditions or phenotypes. Gene annotations and pathway information are often retrieved from databases such as Gene Ontology (GO)[2] or Reactome[3]. Established tools such as GSEA[4], g:Profiler[5], and Enrichr[6] are widely used for pathway enrichment analysis in basic and biomedical research.

Combining multiple omics datasets for gene and pathway analyses is highly beneficial since different data modalities provide complementary biological insights. For instance, transcriptomics and proteomics experiments measure gene and protein expression, post-translational modifications, and signalling network activity. Genomic and epigenomic methods, on the other hand, help us understand genetic variation and gene regulation. Major projects such as the Cancer Genome Atlas (TCGA)[7], Encyclopedia of DNA Elements (ENCODE)[8], Genotype-Tissue Expression project (GTEx)[9], and Clinical Proteomic Tumour Analysis Consortium (CPTAC)[10] provide multidimensional molecular profiles of human tissues, disease states, and cancer samples. Integrative analyses of multi-omics datasets can lead to biological insights, experimental validation, and translational impact.

[1]Computational Biology Program, Ontario Institute for Cancer Research, 661 University Ave Suite 510, Toronto ON M5G 0A3, Canada. [2]Department of Medical Biophysics, University of Toronto, 101 College Str Suite 15-701, Toronto ON M5G 1L7, Canada. [3]Department of Molecular Genetics, University of Toronto, 1 King's College Circle Room 4386, Toronto ON M5S 1A8, Canada. [4]Program in Genetics and Genome Biology, the Hospital for Sick Children Research Institute, 686 Bay Str, Toronto ON M5G 0A4, Canada. [5]Departments of Statistical Sciences, Computer Science and Division of Biostatistics, University of Toronto, 700 University Avenue, Toronto ON M5G 1Z5, Canada. [6]These authors contributed equally: Mykhaylo Slobodyanyuk, Alexander T. Bahcheli. ✉ e-mail: juri.reimand@utoronto.ca

Multi-omics analysis presents unique challenges as omics platforms measure various molecules, have distinct experimental and technical biases, and require specific data processing methods[11]. Comparing genes, transcripts, and proteins directly across the datasets is therefore problematic. We can map omics signals to a common space of pathways and processes to address this complexity[1]. One powerful approach involves data fusion of statistical significance estimates, such as P-values, that effectively accounts for platform-specific confounding effects, assuming appropriate statistical analyses have been performed upstream. Several computational methods are available for this type of analysis[12–18]. Pathway-level methods evaluate pathway enrichments in input omics datasets and integrate these as multi-omics summaries[13,14] while gene-level integration methods prioritise genes or proteins across input datasets and then detect multi-omics pathway enrichments[15–18]. We recently developed the ActivePathways method that first prioritises genes through multi-omics data fusion and then identifies enriched pathways with gene-level evidence from input datasets[18].

Multi-omics datasets often have directional associations, yet these are commonly not considered in integrative analyses. Directional associations may arise from core aspects of cellular logic or experimental design. For example, mRNA and protein expression levels of genes often correlate positively based on the central dogma. Similarly, DNA methylation of gene promoters as a repressive epigenetic mechanism often correlates with lower gene expression. As an example of experimental design, transcriptomic profiles derived from knockout and overexpression experiments of a gene of interest have inverse associations of gene expression changes. While cellular control mechanisms like post-transcriptional or post-translational regulation confound such directional associations, these additional effects are often not measured. Nonetheless, considering directional dependencies in multi-omics data analysis allows researchers to test more specific hypotheses, prioritise genes and pathways with greater accuracy, reduce false-positive findings, and gain detailed mechanistic insights. Currently, directional methods designed for multi-omics data analysis are lacking, leaving an opportunity for the development of such approaches to enhance our understanding of complex biological processes.

Here we propose directional P-value merging (DPM) for directional integration of genes and pathways across multi-omics datasets. DPM employs user-defined directional constraints to prioritise genes or proteins whose directions across omics datasets are consistent with the constraints while penalising those with inconsistent directions. We demonstrate our framework in three case studies: identifying the downstream targets of an oncogenic lncRNA based on transcriptomic profiles from functional experiments in cancer cells; integrating transcriptomic and proteomic data with patient clinical information for cancer biomarker discovery; and characterising *IDH*-mutant subtype of glioma by integrating epigenetic, transcriptomic, and proteomic data. DPM is available in the ActivePathways R package[18] in CRAN.

## Results
### Directional integration of multi-omics data
We developed directional P-value merging (DPM), a statistical method for multi-omics data fusion that prioritises genes across multiple omics datasets by integrating their P-values and directional changes such as fold-changes (FC) (Fig. 1A, Supplementary Fig. 1, Methods). DPM implements a user-defined *constraints vector* (CV) to specify directional associations between input datasets. For each gene, DPM computes a score based on the P-values and directional changes from the omics datasets. Genes showing significant directional changes that comply with the CV are prioritised, while the genes with significant but conflicting directional changes are penalised. DPM builds on our ActivePathways method[18] and provides a directional extension of the

empirical Brown's P-value merging method[19,20]. For a given gene, a directionally weighted score $X_{DPM}$ is computed across $k$ datasets as

$$X_{DPM} = -2(-|\Sigma_{i=1}^{j}\ln(P_i)o_ie_i| + \Sigma_{i=j+1}^{k}\ln(P_i)). \tag{1}$$

To incorporate directionality to P-value merging, we compute sums of log-transformed P-values $P_i$ that are weighted by directional information. Here, $o_i$ shows the observed directional change of the gene in dataset $i$. For example, in differential expression analysis, $o_i$ is the gene fold-change direction relative to a control condition. Directions are considered as unit signs (*i.e.*, $+1$ or $−1$) because effect sizes are generally not comparable between various omics datasets. Besides log-FC values, directions may include correlation coefficients, log-transformed hazard ratio (HR) values from survival analyses, or other values used as unit signs. To obtain $X_{DPM}$, the scores are multiplied by two in line with Fisher's method[21].

The constraints vector CV defines the directional association $e_i$ showing how the direction of dataset $i$ is expected to interact with other input datasets. CV defines the structure of the multi-omics analysis. Series of positive ($+1$) or negative ($−1$) values prioritise genes that have the same observed directions in corresponding datasets (*e.g.*, transcript and protein expression). In contrast, mixed values in CV ($+1$ and $−1$) prioritise genes with inverse directions in corresponding datasets (*e.g.*, DNA methylation and transcript expression). The absolute function in the $X_{DPM}$ formula ensures that CV is globally sign invariant (*i.e.*, $[−1, +1] ≡ [+1, −1]$ and $[+1, +1] ≡ [−1, −1]$): the CV $[+1, +1]$ prioritises genes with up-regulation or down-regulation in both datasets and the CV $[−1, −1]$ results in an equivalent analysis. In contrast, the CVs $[+1, −1]$ and $[−1, +1]$ prioritise genes upregulated in one dataset and downregulated in the other dataset. Importantly, the CV is not limited to the central dogma or any other cellular logic. As a user-defined parameter, it can be configured to highlight genes and pathways with arbitrary directional relationships. An example of data integration with DPM is shown in Supplementary Fig. 1.

DPM can jointly analyse directional and directionless omics datasets. $X_{DPM}$ adds scores over datasets ($1 \dots j$) with directional information and datasets ($j+1 \dots k$) lacking directional information. Either part of the sum can be omitted if needed. In directionless datasets, genes or proteins are only scored based on P-values and are encoded as zeroes in the CV. For example, this can be used for mutational burden tests, epigenetic annotations, or network topology analyses that provide P-values but no directional information.

We compute the merged P-value $P'_{DPM}$ to reflect the joint significance of the gene across the input datasets given directional information. The merged P-value is derived from the cumulative $\chi^2$ distribution as $P'_{DPM} = 1 - \chi^2(\frac{1}{c}X_{DPM}, k')$. For more accurate significance estimation, we account for gene-to-gene covariation in omics data and estimate degrees of freedom $k'$ and scaling factor $c$ from the input P-values using the empirical Brown's method[20]. In addition to DPM, we also provide directional extensions to P-value merging methods by Stouffer[22] and Strube[23] based on the METAL method for genome-wide association studies[24]. We adapted METAL for joint analyses of directional and non-directional multi-omics datasets (Methods).

Our workflow includes four major steps. First, we process upstream omics datasets into a matrix of gene P-values and another matrix of gene directions (Fig. 1A). Dedicated upstream processing of input omics datasets is required to obtain these values. We define a CV with directional constraints based on the overarching hypothesis, experimental design, or biological insights. We also collect up-to-date pathway information[25] from databases such as GO[2] and Reactome[3]. Other types of functional gene sets such as disease genes or transcription factor targets can be used as well. Second, P-values and directions are merged into a single gene list of P-values using DPM or related methods[22,23]. This is useful for multi-omics gene prioritisation.

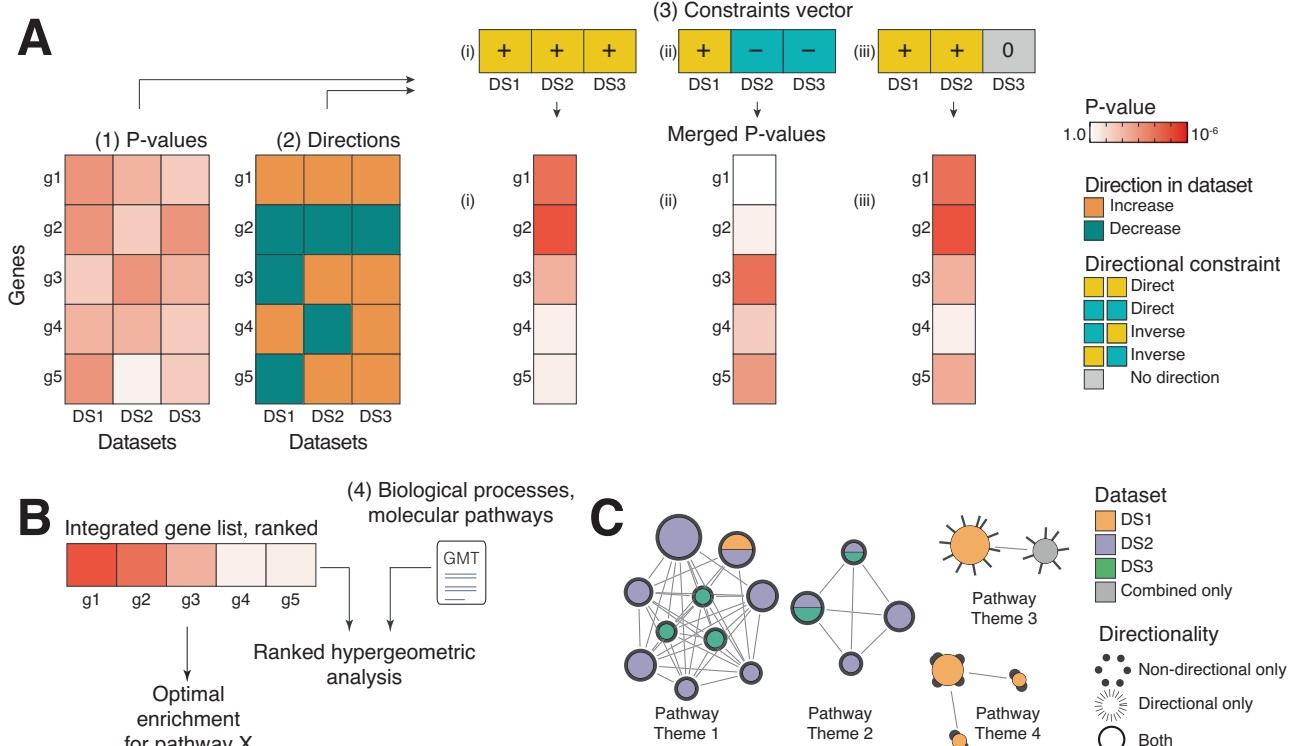

**Fig. 1 | Directional integration of multi-omics data using DPM. A** The DPM method combines gene significance and directions in multi-omics datasets for gene prioritisation and pathway analysis. Four inputs are required: (1) gene activities in input omics datasets quantified as P-values; (2) directional changes of genes such as fold-change (FC) values, used as positive ( + 1) or negative ( − 1) unit values, or zeroes for directionless data; (3) user-defined constraints vector (CV) showing expected directional relationships between the omics datasets; and (4) gene sets of biological processes, pathways, or gene annotations. DPM combines gene P-values and directions with the CV using a data fusion approach, prioritising genes whose directions significantly agree with the CV and penalising those whose directions are inconsistent with the CV. Three examples of CVs are shown. **B** The integrated gene list is analysed for pathway enrichments using ranked hypergeometric tests in ActivePathways to identify the strongest pathway enrichments in top fractions of the ranked gene list and evaluate evidence from input datasets. **C** Enriched pathways are visualised as an enrichment map. The network shows enriched pathways where edges connect pathways that share many genes. Colours indicate the omics datasets that contribute most to pathway enrichments. Node outlines indicate pathways identified using directional or non-directional analyses.

Third, the merged gene list is analysed for enriched pathways using a ranked hypergeometric algorithm in the ActivePathways method[18] that also determines which input omics datasets contribute most to individual pathways (Fig. 1B). Finally, the resulting pathways are visualised as enrichment maps[1,26] that reveal characteristic functional themes and highlight their directional evidence from omics datasets (Fig. 1C). DPM provides a general and adaptable framework to explore understudied intersections of complex multi-omics datasets.

### Benchmarking directional P-value merging

We evaluated DPM and the modified Strube's method using synthetic data (Fig. 2A, B, Supplementary Data 1). Two input datasets of 10,000 genes were integrated in three directional configurations having all genes in directional agreement, all genes in directional conflict, or 50% genes in directional conflict. First, we simulated uniformly distributed P-values as negative controls to evaluate false positive rates of DPM. We tested two scenarios where the two sets of input P-values were either independent (Pearson $r < 0.001$) or strongly correlated with each other ($r = 0.97$). With full directional agreement, DPM expectedly found ~5% of merged P-values at ($P < 0.05$) in independent and correlated datasets, corresponding to the expected fraction of significant P-values in uniform data. This indicates a favourable false positive rate. As directional penalties were applied in DPM, the dataset with 50% directional conflicts showed proportionally fewer significant merged P-values. In contrast, the Strube method found two-fold more significant merged P-values when merging independent P-values

suggesting a higher false positive rate while merging of correlated P-values was not inflated.

Next, we integrated two independent omics datasets having significant signals. We simulated both input datasets using exponentially distributed P-values such that many significant genes were included ( ~ 26% at $P < 0.05$ or ~1% at FDR < 0.05). When all genes were in directional agreement, 39% of merged P-values were significant ($P < 0.05$). This higher fraction is expected as the two input datasets independently contributed to merging. With 50% directional conflicts, 22% of genes were found significant, indicating the role of directional penalties. Even with full directional conflicts, a small fraction of genes (5%) was found significant ($P < 0.05$). Further study of this subset indicated that DPM prioritised directional conflicts where the gene was supported by strong significance in one dataset while the directionally conflicted evidence from the second dataset was not significant (Fig. 2B). This suggests increased sensitivity of DPM towards weaker effects. The modified Strube method again showed a consistently higher rate of significant findings, suggesting an inflation in merging independent P-values.

Finally, we integrated two correlated omics experiments having significant signals. We simulated exponentially distributed P-values with a large fraction of significant genes that were highly correlated between the two datasets ($r = 0.97$). DPM found fewer significant merged P-values compared to independent datasets. This is expected as DPM adjusts for covariation of input P-values for more conservative merging. DPM and the Strube method behaved similarly in integrating correlated datasets. In both cases, no significant results were found

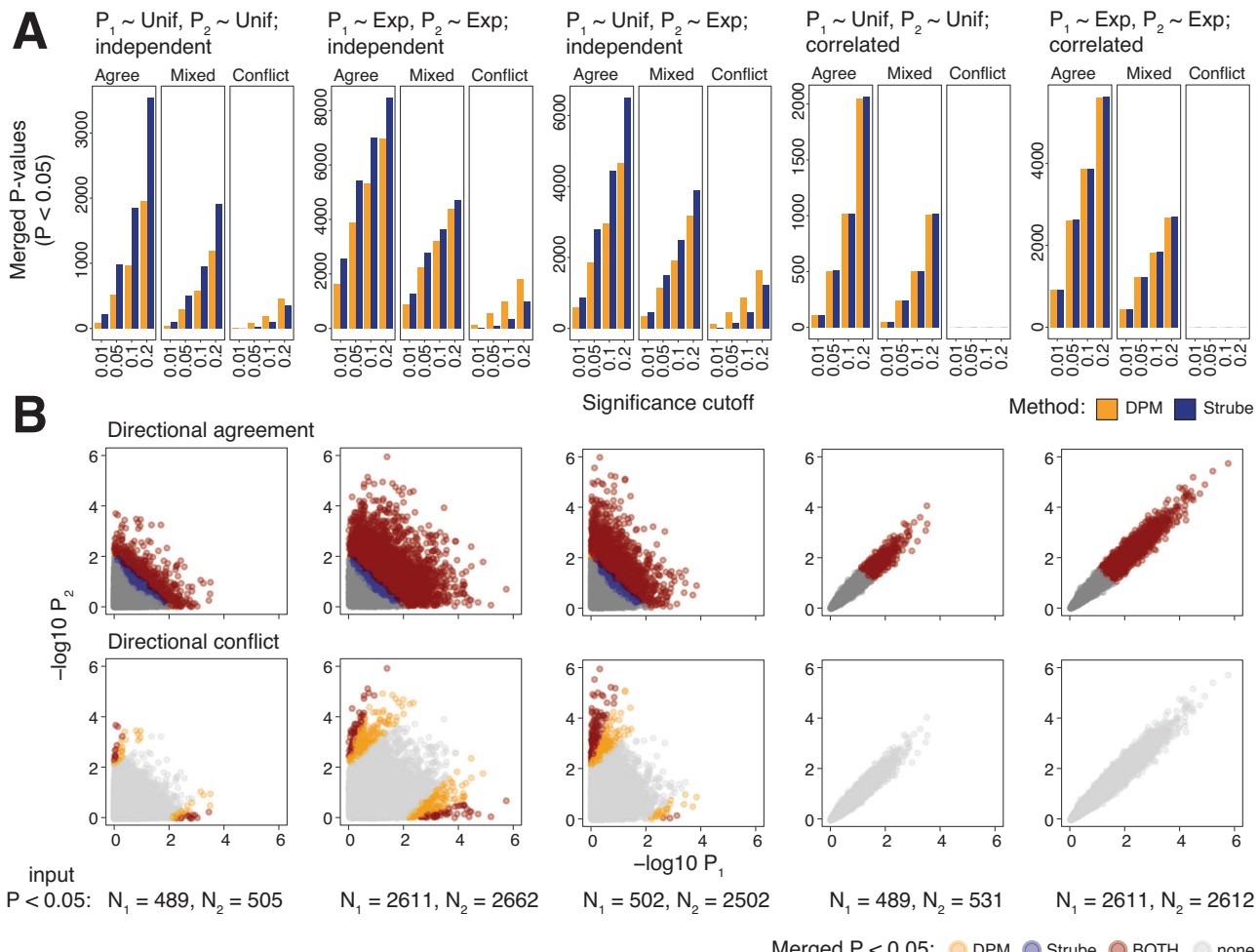

**Fig. 2 | Evaluating directional P-value merging (DPM) with simulated data.** Two sets of 10,000 genes with simulated P-values and directional information were merged using DPM and the modified Strube method. Input P-values $P_1$ and $P_2$ were generated randomly from the uniform distribution (Uni) as negative controls, or from the exponential distribution (Exp) to reflect datasets with significant signals. Input P-values were generated independently or with strong correlations. Three types of directions were considered: directional agreement of all genes, directional conflict of all genes, and 50/50 mixed directions. Unadjusted P-values are shown. **A** Bar plots show significant merged P-values at various cut-offs. DPM finds the expected fraction of significant results in uniformly sampled data while Strube's method shows inflated results when merging independent P-values. **B** Scatter plots show the distributions of input P-values. Points are coloured based on merged significance from the two methods ($P < 0.05$). $N_1$ and $N_2$ show numbers of significant input P-values. Scatter plots suggest that DPM is more sensitive in directionally integrating genes in which the directional conflicts are not supported by significant P-values (yellow points).

when all genes were in conflict, indicating that directional penalties were stronger in highly correlated input datasets. These benchmarks suggest that DPM is a statistically well-calibrated approach for directional integration of multi-omics data.

## Analysing transcriptomic targets of *HOXA10-AS* lncRNA in glioma

We then studied real omics datasets using DPM. First, we analysed an earlier transcriptomics dataset in which the oncogenic lncRNA *HOXA10-AS* was profiled in knockdown (KD) or overexpression (OE) experiments in patient-derived glioblastoma (GBM) cells[27]. To identify target genes and pathways of the lncRNA, we prioritised genes that changed in opposite directions in the two experiments and penalised genes that were either upregulated or downregulated in both experiments (Fig. 3A). DPM revealed 2236 significant and directionally consistent genes ($P < 0.05$) (Fig. 3B, Supplementary Data 2). Further, we found 773 genes that were penalised by DPM due to directional constraints, however these were identified in the reference non-directional analysis ($P < 0.05$). Among prioritised genes, *CPED1* was a top result found by DPM ($P = 2.8 \times 10^{-7}$). *CPED1* was significantly upregulated in

*HOXA10-AS* KD experiment and downregulated upon OE (Fig. 3C), indicating a potential negative regulatory target of *HOXA10-AS*. *CPED1* is a little-studied gene that encodes a cadherin-like protein with a PC-esterase domain. Also, the tumour suppressor gene *FAT1* was prioritised due to upregulation in *HOXA10-AS* OE and no significant change in KD, exemplifying another mode of gene prioritisation in DPM. *FAT1* encodes a cadherin protein and tumor suppressor that controls organ growth, cell polarisation, and cell-cell contacts and is involved in tumor invasion, metastasis, and drug resistance[28,29]. In contrast, the top directionally penalised genes included *NEGR1*, a neuronal growth regulator, and *CACNA1H*, a calcium voltage-gated channel, that were either jointly upregulated or jointly downregulated in KD and OE experiments (Fig. 3C). *NEGR1* and *CACNA1H* are involved in neuronal development and cell adhesion, respectively[30,31].

Directional pathway analysis using DPM revealed 138 enriched GO processes and Reactome pathways (ActivePathways with DPM, family-wise error rate (FWER) < 0.05) (Fig. 3D, E, Supplementary Data 3 and 4). The reference non-directional analysis found 219 pathways and processes (ActivePathways with Brown, FWER < 0.05). Six pathways were only found by DPM through directional information: vesicular

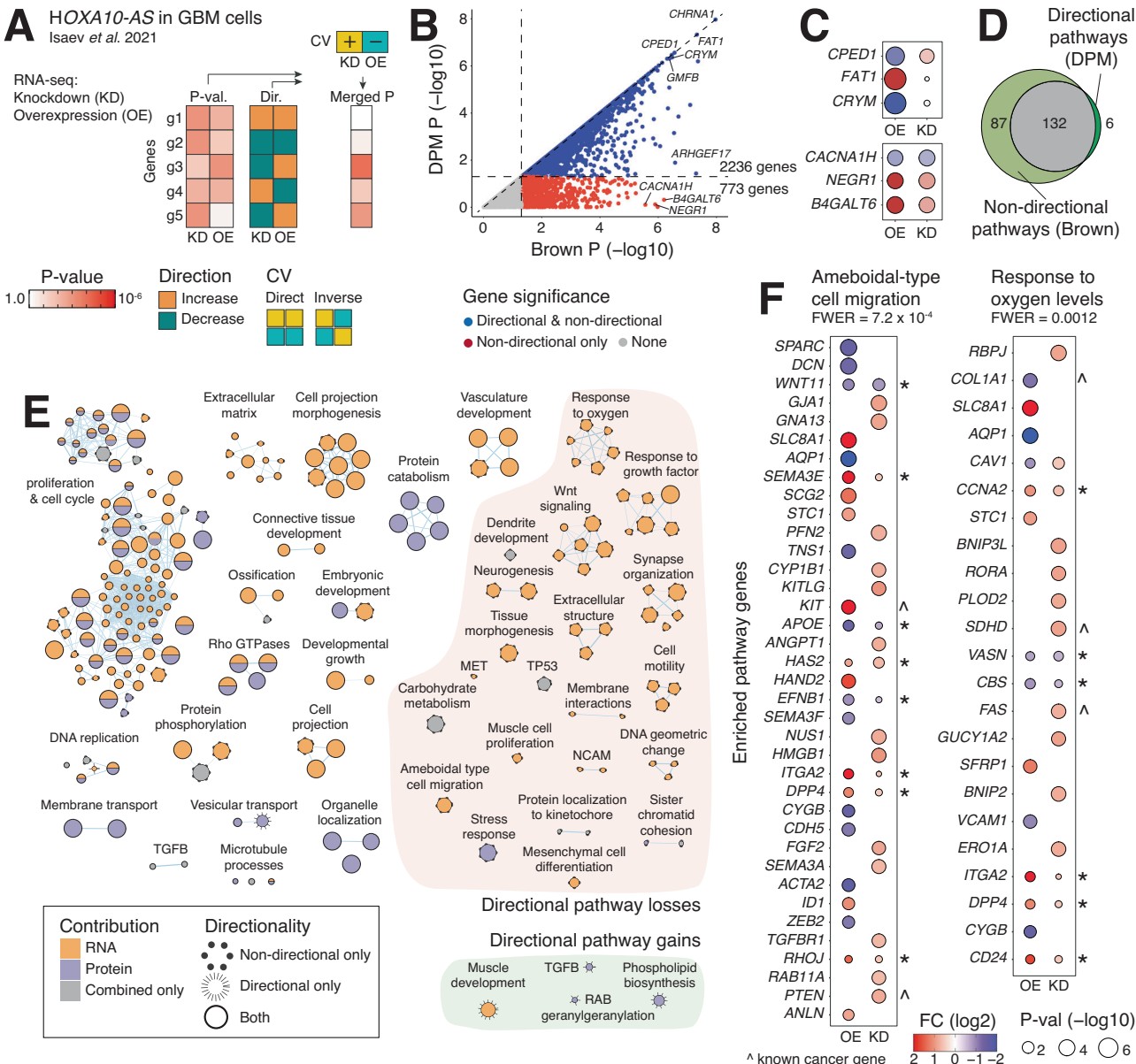

**Fig. 3 | Directional integration of transcriptomics data from functional experiments of *HOXA10-AS* lncRNA in GBM cells. A** We integrated differential gene expression data from *HOXA10-AS* knockdown (KD) and overexpression (OE) experiments from a previous study[27] that compared sets of three replicates. DPM prioritised genes that showed different fold-change (FC) directions in KD and OE experiments and penalised genes with matching directions using the constraints vector (CV) [KD = −1, OE = +1]. **B** Scatter plot of merged P-values from directional analysis (DPM, Y-axis) and non-directional analysis (the Brown method, X-axis). Prioritised genes with directionally consistent changes are shown on the diagonal or closely below it (blue), while directionally penalised genes with conflicting directional changes are further below the diagonal (red). Unadjusted P-values are shown. **C** Examples of prioritised genes (top) and penalised genes (bottom). **D** Venn diagram of enriched pathways found with directional and non-directional analyses

(family-wise error rate (FWER) < 0.05). **E** Enrichment map of pathways and processes from directional and non-directional analyses (FWER < 0.05). Pathways are shown as node in the network that are connected by edges if the pathways share many genes. Subnetworks represent functional themes. Node colours indicate dataset contributions (KD, OE, both, or combined-only). Node size reflects number of genes per pathway. Node outlines show directionally prioritised pathways (spiky edges), directionally penalised pathways (dotted edges), or pathways found using both approaches (solid edges). Major groups of directionally prioritised or penalised pathways are grouped on the right. **F** Dot plots of significant genes involved in cell migration and oxygen response processes visualised with P-values and fold-change values from the *HOXA10-AS* transcriptomics study[27]. Genes penalised in the non-directional analysis are indicated with asterisks. Carets show known cancer genes from the COSMIC Cancer Gene Census database[53].

transport, RAB geranylgeranylation, TGF-beta signalling, muscle development, DNA replication, and phospholipid biosynthesis. On the other hand, a third of the enriched pathways from the non-directional analysis (87/219), including cell motility, brain development, and oxygen response, were excluded by DPM due to directional disagreements in related genes such as *DPP4*, *STC1*, and *ADGRL2* (Supplementary Fig. 2). Although these processes are central to glioma biology[32–34],

our analysis suggests that these are not directly regulated by *HOXA10-AS* since related genes often showed directional conflicts in KD and OE experiments. For example, the GO process *ameboidal-type cell migration* found in the non-directional analysis included 37 differentially expressed genes (FWER = 7.3 × 10$^{-4}$). Eight genes were directionally inconsistent due to either upregulation or downregulation in both experiments (*WNT11*, *SEMA3E*, *APOE*, *HAS2*, *EFNB1*, *ITGA2*, *DPP4*,

*RHOJ*) (Fig. 3F). Penalising these genes directionally led to loss of pathway enrichment. Similarly, four oxygen-related processes were lost, such as the GO process *response to oxygen levels* (FWER = 0.0012), in which directional conflicts occurred in 6 of 23 enriched genes (Fig. 3F).

This analysis demonstrates the integration of transcriptomic data from two functional experiments on a target gene of interest. We expect that genes and pathways with opposite directional changes in KD and OE experiments are regulated by *HOXA10-AS*, an oncogenic lncRNA in glioma[27]. On the other hand, genes and pathways that are unidirectionally regulated in KD and OE experiments may respond to *HOXA10-AS* levels through feedback loops or post-transcriptional regulation or alternatively reflect a broader cellular response downstream of *HOXA10-AS*. We can prioritise such genes and pathways using an alternative CV that prioritises matching gene directions (Supplementary Fig. 3). Integrating directional associations from functional experiments improves the resolution of gene prioritisation and pathway enrichment analysis.

## Proteogenomic analysis of ovarian cancer for biomarker discovery

Next, we integrated cancer transcriptomics and proteomics data with patient overall survival (OS) in ten cancer types from the CPTAC project[10] (Fig. 4A, Supplementary Fig. 4, Supplementary Data 5). First, we asked which genes significantly associated with OS via transcript or protein expression using Cox proportional-hazards (PH) regression using patient age, sex, and tumor stage as covariates. P-values and hazard ratios (HR) for transcript- and protein-level OS associations were integrated using DPM such that genes with consistent OS associations were prioritised while inconsistent associations were penalised.

We focused on the ovarian cancer dataset (OV) with 169 serous cystadenocarcinoma samples. DPM identified 907 significant genes ($P_{DPM} < 0.05$). 192 genes were penalised due to inconsistent survival associations compared to a reference non-directional analysis ($P_{Brown} < 0.05$) (Fig. 4B, Supplementary Data 6). Directionally prioritised genes had consistently positive or negative OS associations with protein and transcript expression, while penalised genes showed mixed OS associations (Fig. 4C). The top prioritised gene *ACTN4* ($P_{DPM} = 5.4 \times 10^{-9}$) encodes a cytoskeletal actin-binding protein and an emerging oncogene linked to poor prognosis in ovarian cancer[35]. Higher transcript and protein expression of *ACTN4* associated with worse prognosis in OV (Fig. 4D), and mRNA and protein levels of *ACTN4* were highly correlated (Spearman $\rho = 0.75$, $P < 2.2 \times 10^{-16}$) (Fig. 4E). In contrast, the top penalised gene *PIK3R4* showed inconsistent OS associations: higher transcript expression associated with worse prognosis while higher protein expression associated with improved prognosis, and transcript and protein expression levels were not correlated (Fig. 4D-E). *PIK3R4* encodes a regulatory kinase subunit in the PI3K/AKT pathway, a central signalling network that controls cancer cell proliferation, survival, and metabolism[36,37]. Inconsistent survival associations of *PIK3R4* expression suggest additional modes of regulation that remain masked in these transcriptomics and proteomics datasets.

Pathway analysis with DPM revealed 170 significant pathways and processes with multi-omics survival associations (ActivePathways FDR < 0.05), including major functional themes of proliferation, focal adhesion, cell motility, immune cell activity, and development, and signalling pathways such as Hedgehog, Notch, and NFKB (Fig. 4F, G, Supplementary Data 7 and 8). Compared to the reference non-directional analysis, DPM penalised multiple pathways due to directional conflicts in OS associations with transcript and protein expression. For example, biological processes of protein translation and degradation, RNA modifications, and mitochondrial function were penalised, in line with previous reports that indicated low correlations of transcript and protein expression levels in such genes[38–40]. For

example, the GO process *mitochondrial translation* was identified in the non-directional analysis; however, it was penalised in the directional analysis since several enriched pathway genes (8/33) had inconsistent OS associations with transcript and protein expression (Fig. 4H). This analysis demonstrates the integration of multi-omics datasets with clinical information to discover biomarkers and biological mechanisms in heterogeneous datasets of patient cancer samples.

## Integrating multi-omics data to study *IDH*-mutant glioma

Lastly, we compared glioma samples based on the mutation status of isocitrate dehydrogenase 1 (*IDH1*), a well-established molecular marker of glioma that indicates lower-risk disease[41]. We integrated DNA methylation, transcriptomics, and proteomics datasets from TCGA and CPTAC by modelling positive and negative directional associations between the three data types (Fig. 5A). DNA methylation of gene promoters is a repressive epigenetic mechanism that often correlates with reduced gene expression; therefore, we can obtain more accurate multi-omics maps by inversely associating methylation with gene expression. First, we analysed differential transcript and protein expression and DNA promoter methylation in *IDH*-mutant GBMs relative to *IDH*-wildtype GBMs and found hundreds of significant genes (Fig. 5B). However, only few genes (32) were significantly detected across all three datasets, and even fewer consistently up-regulated and down-regulated genes were found.

To study the molecular makeup of *IDH*-mutant gliomas in greater detail, we analysed the multi-omics dataset directionally by prioritising inverse associations of promoter methylation levels with direct associations of protein and transcript levels (Fig. 5A). DPM analysis revealed 2023 significant genes ($P < 0.05$; Fig. 5C, Supplementary Data 9). In addition, 267 genes were penalised due to directional conflicts compared to the reference non-directional analysis (Brown, $P < 0.05$). Directionally prioritised genes were often driven by elevated promoter methylation and reduced transcript and protein expression that is consistent with the hypermethylator phenotype of *IDH*-mutant gliomas[42]. In contrast, the genes penalised by DPM often showed elevated promoter methylation combined with gene upregulation at transcript or protein level (Fig. 5D), potentially due to additional epigenetic regulation that is not measured in our data. We found 98 known cancer-associated genes using DPM (FDR < 0.05), of which 26 (27%) were consistently regulated between the three datasets. Pathway enrichment analysis of directionally prioritised genes revealed 72 pathways and processes (FWER < 0.05, ActivePathways), while 33 pathways from the non-directional reference analysis were penalised by DPM (Fig. 5E, Supplementary Data 10 and 11). DPM penalised biological processes and pathways that appear to be less relevant to glioma biology. For example, the GO process *muscle organ development* was found in the non-directional analysis, however it was penalised by DPM due to directional conflicts in 80 of 195 genes (Fig. 5F). Fibroblast growth factor receptor (FGFR) signalling pathways were also penalised in the directional analysis (Supplementary Fig. 5), such as the GO process *negative regulation of fibroblast growth factor receptor signalling pathway* that included ten genes in the non-directional analysis. However, three genes *FGF2*, *WNT5A*, and *SULF1* were penalised due to directional conflicts in increased promoter methylation coupled with higher gene expression. FGFR signalling regulates tumor progression in gliomas[43,44] and oncogenic alterations of FGFR genes have been found in *IDH*-wildtype gliomas, such as *FGFR-TACC* fusions in GBM[45] and structural variants of *FGFR1* in pediatric gliomas[46]. However, our analysis was focused on *IDH*-mutant gliomas and indicated inconsistent regulation of FGFR-related genes.

Encouragingly, some processes such as *gliogenesis* were only found in the directional analysis as several related genes showed significant and directionally consistent changes in *IDH*-mutant gliomas (FWER = 0.0207) (Fig. 5G). For example, *OLIG2* was upregulated in *IDH*-

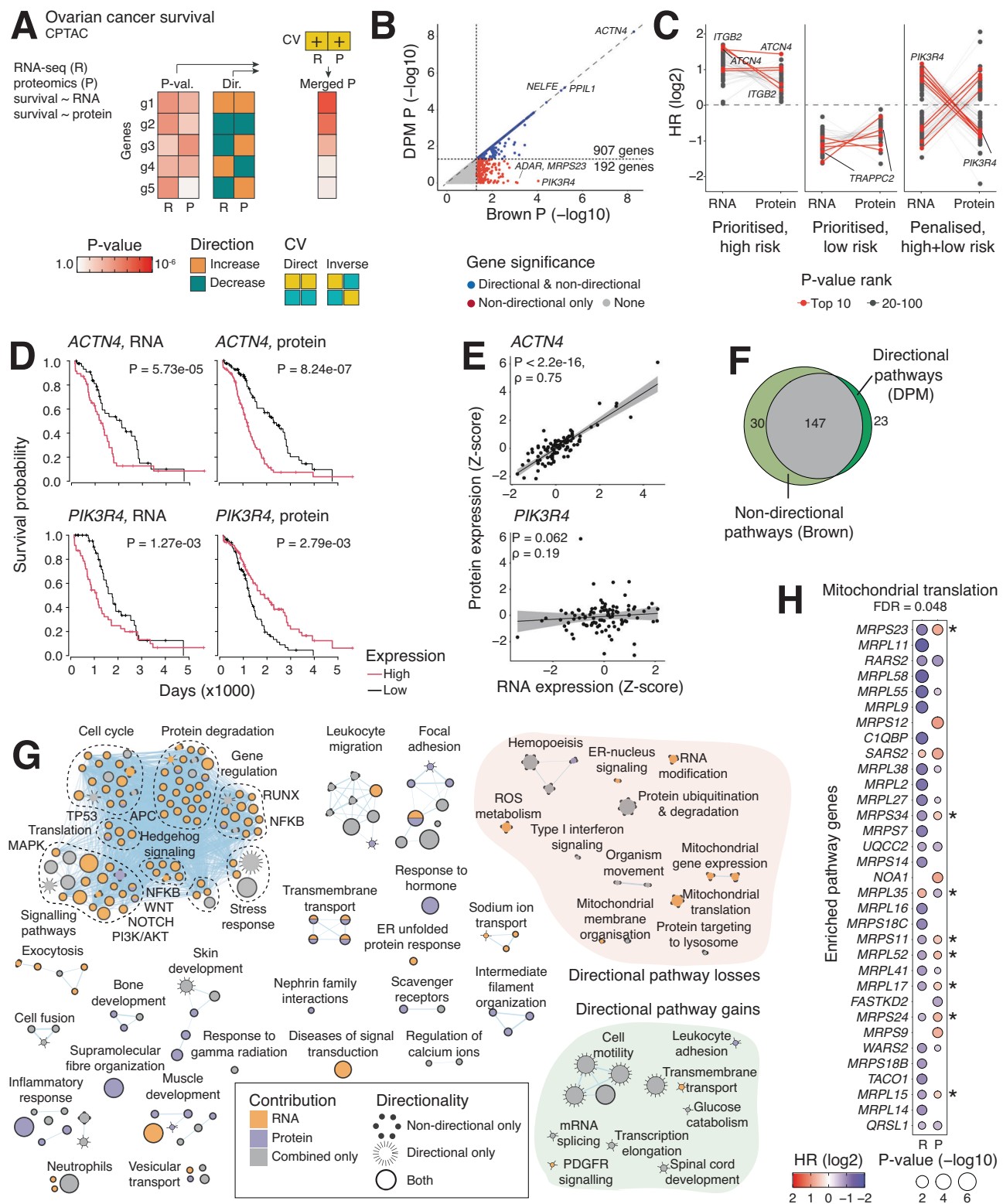

mutant gliomas at the mRNA and protein level. *OLIG2* encodes a core neurodevelopmental transcription factor that controls a stem-like tumor-propagating cell state in GBM[47].

Finally, we validated our analysis of *IDH*-mutant gliomas in an independent set of cancer samples. We integrated promoter methylation and gene and protein expression datasets from the GLASS project[48] and the proteogenomics dataset by Oh et al.[49] (Fig. 5H, Supplementary Data 12–14). Directional analysis revealed 170 significant pathways in the validation dataset (FWER < 0.05,

Supplementary Fig. 6). Major functional themes such as cell adhesion, cell motility, hypoxia, apoptosis, and cell proliferation were found in both datasets. The validation dataset revealed additional processes of immune system, MAPK signalling, and others, while a few cell differentiation and growth factor signalling pathways were only found in the discovery dataset. This pathway-level validation in an independent set of glioma samples lends confidence to our method and demonstrates data integration through diverse clinical multi-omics datasets.

**Fig. 4 | Integrating ovarian cancer transcriptomes and proteomes with patient survival information for pathway and biomarker analyses. A** We correlated mRNA (R) and protein (P) levels for each gene with patient overall survival (OS) in 169 ovarian serous cystadenocarcinoma (OV) samples using clinical covariates (patient age, patient sex, tumor stage) in Cox proportional-hazards (PH) models. We prioritised genes that showed matching OS associations with mRNA and protein levels and penalised genes with opposite OS associations using the constraints vector (CV) [R = +1, P = +1]. Unadjusted chi-square P-values and hazard ratio (HR) values from Cox-PH models were used for directional data integration and are shown in panels C, D, and H. **B** Scatter plot of merged P-values of OS associations in OV from directional analysis (DPM, Y-axis) and non-directional analysis (Brown, X-axis). Prioritised genes with consistent OS associations are shown on the diagonal or closely below it (blue), while directionally penalised genes are further below the diagonal (red). Unadjusted P-values are shown. **C** Log-transformed HR values of top 100 genes prioritised or penalised by DPM. Prioritised genes associate with either higher or lower risk at mRNA and protein levels, while penalised genes have mixed risk associations with mRNA and protein expression. **D** Kaplan-Meier plots of OS

associations of top genes. High mRNA and high protein levels of the top prioritised gene *ACTN4* associate with worse prognosis. In contrast, mRNA and protein levels of the top penalised gene *PIK3R4* show inverse OS associations. **E** Scatterplots of mRNA and protein expression of *ACTN4* and *PIK3R4*. Spearman correlation coefficients and P-values from two-sided correlation tests are shown. Correlation trendline is shown with 95% confidence intervals. **F** Venn diagram of enriched pathways of OS associations with mRNA and protein levels from directional and non-directional analyses (ActivePathways, false discovery rate (FDR) < 0.05). **G** Enrichment map of pathways and processes with OS associations. The network shows pathways as nodes that are connected by edges and grouped into functional themes if the corresponding pathways share many genes. Major groups of directionally prioritised or penalised pathways are grouped on the right. **H** Dot plot of significant genes involved in mitochondrial translation. This process was penalised in the directional analysis due to several genes showing inconsistent OS associations with mRNA and protein expression. Asterisks show directionally penalised genes.

## Discussion

We describe a data fusion algorithm for directional gene prioritisation and pathway enrichment analysis in multi-omics datasets using directional constraints. The method is broadly applicable to various analytical workflows and experimental designs as it relies only on appropriately derived P-values and directional changes of genes. To demonstrate our method, we analyse multi-omics datasets from cancer cell lines and heterogeneous patient cohorts. We encode various directional constraints to capture complex interactions of genes and pathways in omics datasets. We also integrate patient clinical information to discover candidate biomarkers and explore the molecular phenotypes of high-risk disease. We validate our method by recovering pathways and processes characteristic of *IDH*-mutant gliomas in an independent set of cancer samples.

A notable limitation of our approach is that directional constraints only provide a simplified representation of cellular logic. For example, transcript and protein levels are sometimes not correlated due to factors that are not measured directly, such as post-translational modifications, protein-protein interactions, alternative splicing, or feedback loops. Limited transcript-protein correlations have been described in the context of protein translation, mRNA splicing, oxidative phosphorylation, electron transport chain, and other housekeeping processes[38–40,50]. Similarly, here we used DNA methylation of gene promoters for simplicity, however distal enhancers also contribute to gene regulation and could be incorporated into directional analyses. However, our method remains valid given the assumptions of directional constraints. Constraints can be adapted in many ways to account for biological complexity and ask specific questions in multi-omics datasets. For example, one can prioritise genes that have inversely associated transcript and protein levels to study additional mechanisms of post-transcriptional control.

Our data fusion framework is broadly applicable as it makes only a few assumptions about input data. Some considerations are noted. First, accurate upstream data processing is an essential requirement. Omics platforms require dedicated data processing methods to identify significant signals and account for biases. Our method relies on accurately computed P-values, which need to be well calibrated and comparable between the input datasets. Second, we only use discrete gene directions represented as unit signs (+1 or −1) that are derived from fold-change values, correlation or regression coefficients, or hazard ratios. Discrete directions are simple and robust and can be extracted easily from case-control comparisons, time series, and clustering. In contrast, numeric directions would be error-prone as these are generally not comparable between omics platforms. Instead, we assume that P-values reflect the strengths of gene directions. Third, genes, proteins, transcripts, sites in non-coding

DNA, and other elements measured in multi-omics datasets need to be mapped to a common namespace of genes. Finally, common limitations of pathway enrichment analysis[1] also apply to our method: for example, pathway analyses tend to include redundant information and introduce biases towards well-studied genes and processes. We envision several areas of future work. Our current method is designed for analysing bulk omics datasets and single-cell datasets in common workflows that integrate across a relatively small number of omics profiles or clusters. More work is needed to ensure the scalability of our method to large numbers of multi-omics profiles. Second, molecular pathways and biological processes are currently collapsed into gene sets, however, similar data fusion methods are needed for molecular interaction networks. In summary, our directional multi-omics analysis enables mechanistic and translational insights by focusing on understudied intersections of complex omics datasets.

## Methods

### Directional P-value merging (DPM)

To integrate multiple omics datasets through gene P-values and directional information, we implemented or repurposed directional extensions to four P-value merging methods by Fisher[21], Brown[19,20], Stouffer[22], and Strube[23]. Methods by Brown and Strube were originally developed to account for the covariation of gene P-values across input datasets based on methods by Fisher and Stouffer, respectively. All methods assume that P-values are uniformly distributed under the null hypothesis and well calibrated. Covariation-adjusted methods account for dependencies in P-value distributions and thereby provide more conservative merged P-values. As omics datasets include biological dependencies, covariation-adjusted methods are usually more appropriate.

Fisher's method takes the null hypothesis that the true effect in each of the combined datasets is zero and the alternative hypothesis that at least one dataset has a non-zero effect. It assumes that independent P-values are used as input. It collapses $k$ P-values $P_i$ to a score $X_F$ based on the sum of log-transformed P-values. The score $X_F$ is transformed into a merged P-value $P'_F$ through the cumulative $\chi^2$ distribution with $2k$ degrees of freedom, as

$$X_F = -2\Sigma_{i=1}^{k}\ln(P_i),\qquad(2)$$

$$P'_F = 1 - \chi^2(X_F, 2k).\qquad(3)$$

Brown's method extends Fisher's method to account for P-value covariation in input datasets by approximating the score

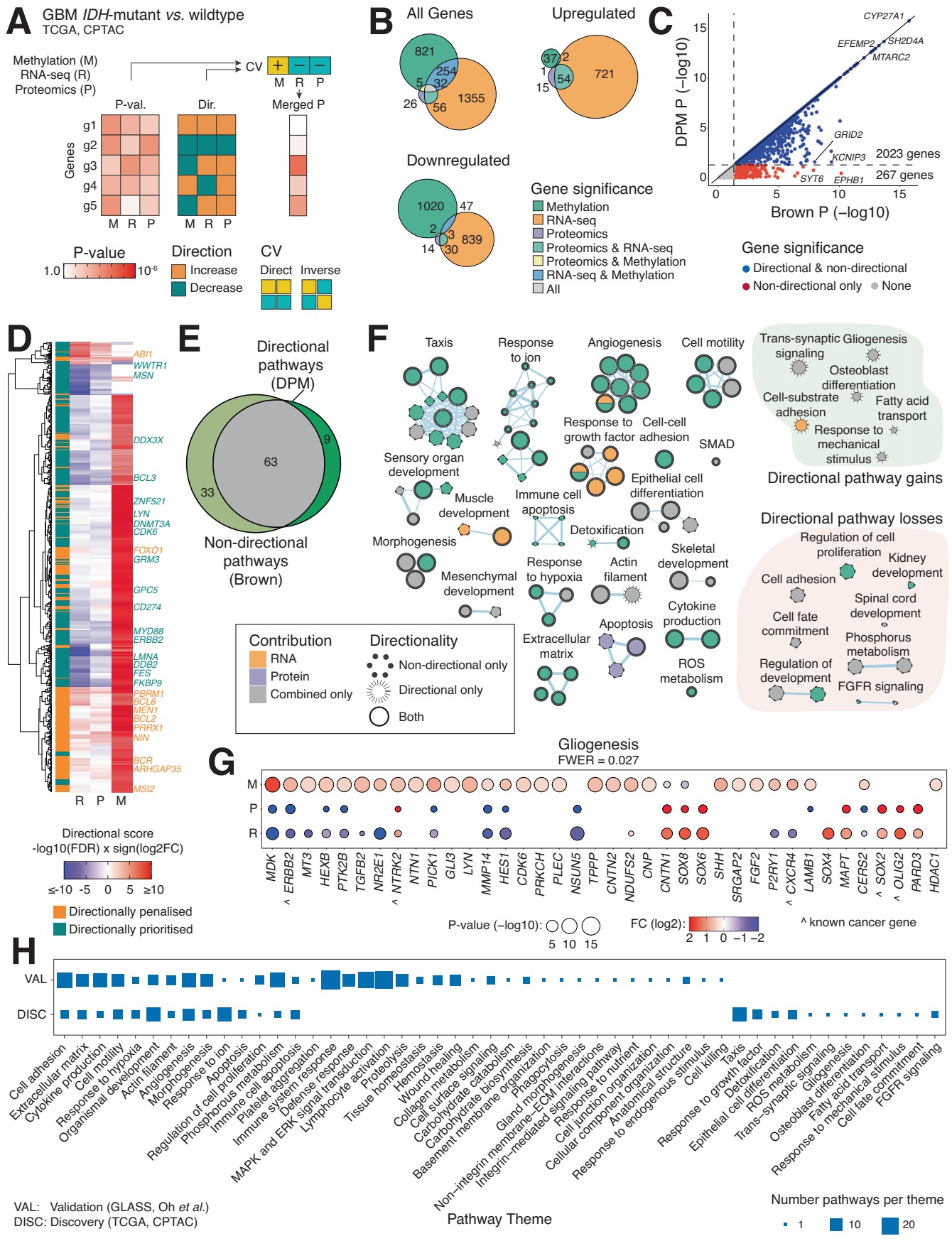

**Fig. 5 | Integrating transcriptomic, proteomic, and DNA methylation profiles of *IDH*-mutant gliomas. A** We compared transcript and protein expression and promoter DNA methylation of *IDH*-mutant and *IDH*-wildtype gliomas. We prioritised mRNA (R) and protein (P) expression levels that directly associated with each other and inversely associated with promoter DNA methylation (M) using the constraints vector (CV) [M = +1, R = −1, P = −1]. At least six *IDH*-mutant and 90 *IDH*-wildtype samples were included depending on data type. **B** Venn diagrams of significant genes found separately in three input datasets (false discovery rate (FDR) < 0.1, Mann-Whitney U-tests). Downregulated genes showed reduced mRNA and protein expression and increased promoter methylation, while upregulated genes showed decreased promoter methylation. **C** Scatter plot of merged P-values from directional analysis (DPM, Y-axis) and non-directional analysis (Brown, X-axis). Prioritised genes with consistent multi-omics directions are shown on the diagonal or closely below it (blue), while directionally penalised genes are further below the diagonal (red). Unadjusted P-values are shown. **D** Heatmap of significantly penalised or prioritised top genes (Brown, FDR < 0.001). Prioritised genes were often

characterised by high promoter methylation and reduced mRNA and protein expression, while penalised genes often showed high promoter methylation and increased expression. Known cancer genes are listed and coloured as directionally penalised or prioritised. **E** Venn diagram of enriched pathways from the directional and non-directional analyses (ActivePathways, family-wise error rate (FWER) < 0.05). **F** Enrichment map of pathways and processes in *IDH*-mutant glioblastoma. The network shows pathways as nodes that are connected by edges if the corresponding pathways share many genes. Major groups of directionally prioritised or penalised pathways are grouped on the right. **G** Dot plot of significant genes involved in the gliogenesis process. This process was only detected in the directional analysis as several related genes showed significant and directionally consistent changes. Unadjusted P-values from Mann-Whitney U-tests are shown. Carets show known cancer genes. **H** Validating the multi-omics analysis of *IDH*-mutant gliomas in an independent dataset. Functional themes from the discovery dataset (TCGA, CPTAC) and validation dataset (GLASS[48], Oh et al.[49]) were compared. Known cancer genes were retrieved from COSMIC Cancer Gene Census[53] (panels **D**, **G**).

$X_F$ from Fisher's method using a scaled $\chi^2$ distribution. Scaling factor $c$ and updated degrees of freedom $k'$ are derived as $c = \frac{\text{Var}[X]}{2E[X]}$ and $k' = \frac{2(E[X])^2}{\text{Var}[X]}$, respectively. The expected value and variance of the scaled distribution are derived as $E[c\chi^2(k')] = ck'$ and $\text{Var}[c\chi^2(k')] = 2c^2k'$, respectively. The merged Brown P-value $P'_B$ is computed based on the sum of log-transformed P-values from the cumulative scaled $\chi^2$ distribution with scaling factor $c$ and degrees of freedom $k'$, as

$$X_B = -2\Sigma_{i=1}^{k}\ln(P_i), \tag{4}$$

$$P'_B = 1 - \chi^2\left(\frac{X_B}{c}, k'\right). \tag{5}$$

Empirical Brown's method (EBM)[20] estimates the expected value and variance from the input datasets nonparametrically. We used EBM here and refer to it as Brown's method.

To incorporate directionality to Fisher's and Brown's methods, our method takes the null hypothesis that the true effect in each of the combined datasets is zero given directional constraints between the datasets and the alternative hypothesis that the effect of at least one dataset is not zero given the directional constraints. We jointly analyse directional information representing the observed gene direction $o_i$ and the expected directional association $e_i$ in each dataset $i$. For example, in differential gene expression analyses of two conditions relative to a control condition, $o_i$ is the sign of fold-change of the gene in condition $i$, and $e_i$ is the expected directional agreement of the two conditions. Both $o_i$ and $e_i$ take values +1, −1 or 0. The constraint vector (CV) [+1, +1] prioritises genes with consistent fold-change directions across the two conditions and is equivalent to the CV [−1, −1]. Alternatively, the CV [+1, −1] and the CV [−1, +1] both prioritise genes with opposite fold-change directions across two conditions. Values of zero are used for both $o_i$ and $e_i$ to define datasets that have no directional information. Directional terms $o_i$ and $e_i$ are incorporated as weights in the sum log-transformed P-values as

$$X_{DPM} = -2\left(-|\Sigma_{i=1}^{j}\ln(P_i)o_ie_i| + \Sigma_{i=j+1}^{k}\ln(P_i)\right). \tag{6}$$

Here, datasets (1, 2, …, $j$) have directional information while datasets ($j$+1, $j$+2, …, $k$) have no directional information. This permits joint analyses of directional and directionless datasets and either part of the sum can be omitted depending on data availability. Intuitively, directional agreements increase the sums of log-transformed P-values and cause increased significance of the resulting merged P-value, while directional disagreements reduce the sums and decrease overall significance. The absolute function ensures that the CV is globally sign

invariant (*i.e.*, [−1, +1] ≡ [+1, −1] and [+1, +1] ≡ [−1, −1]). The overall sum is multiplied by −2 similarly to the methods by Fisher and Brown. Finally, a scaled cumulative $\chi^2$ distribution is computed from Brown's method to obtain the merged P-values directionally as

$$P'_{DPM} = 1 - \chi^2\left(\frac{1}{c}X_{DPM}, k'\right). \tag{7}$$

This method is referred to as DPM (directional P-value merging). An example of this calculation is shown in Supplementary Fig. 1.

In addition to DPM, we implemented a directional extension of the METAL method[24] that extends Stouffer's method[22] for meta-analysis of GWAS studies. Each study has a direction of effect that reflects the impact each allele has on the observed phenotype. This observed directional term, $o_i$, can either be positive (+1), reflecting an increase in the observed phenotype, or negative (−1), reflecting a decrease. Directional Stouffer's method introduced by METAL converts P-values from $k$ independent tests into Z-scores using the inverse of the standard normal cumulative distribution function $\Phi^{-1}$ as

$$Z_M = \frac{\sum_{i=1}^{k}\Phi^{-1}\left(\frac{P_i}{2}\right)o_i}{\sqrt{k}}. \tag{8}$$

Merged P-values are generated through the standard normal cumulative distribution function as $P'_M = 2\Phi(-|Z_M|)$. To account for P-value dependencies, Strube's extension to Stouffer's method[23] leads to more conservative significance estimates by incorporating the overall covariation of P-values in input datasets, similarly to Brown's extension of Fisher's method. We implemented a directional extension of Strube's and Stouffer's methods similarly to METAL as

$$Z_S = \left|\frac{\sum_{i=1}^{j}\Phi^{-1}\left(\frac{P_i}{2}\right)o_ie_i}{\sqrt{j}}\right| + \frac{\sum_{i=j+1}^{k}\Phi^{-1}\left(\frac{P_i}{2}\right)}{\sqrt{k-j}}. \tag{9}$$

Here, Z-scores are acquired for the directional datasets (1, 2, …, $j$) separately from the non-directional datasets ($j$+1, $j$+2, …, $k$) and then each term is combined before calculating a merged P-value, similarly to DPM above.

DPM is available as part of the ActivePathways R package in CRAN (https://cran.r-project.org/web/packages/ActivePathways/) and GitHub (https://github.com/reimandlab/ActivePathways).

## Pathway enrichment analysis

Pathway enrichment analysis is implemented in the ActivePathways R package as described previously[18]. The input to pathway enrichment analysis is a gene list ranked by P-values from directional or non-directional data integration. ActivePathways uses the ranked hypergeometric test to analyse the ranked gene list to determine optimal enrichments of individual gene sets such as biological processes from Gene Ontology[2] and molecular pathways of Reactome[3]. We recommend limiting gene sets by size (e.g., 10-1000 genes by default) to exclude overly generic and too specific gene sets that lead to statistical and interpretative biases. Holm family-wise error rate (FWER)[51] is used for multiple testing correction at the pathway level by default, however the Benjamini-Hochberg false discovery rate (FDR)[52] can be also used for less-stringent corrections. It is important to consider background gene sets for accurate pathway enrichment analyses for cases where only a subset of genes, transcripts, or proteins are measured in an input omics experiment. Best practices of pathway enrichment analysis are described in a recent review paper[1].

## Evaluating DPM using simulated and real datasets

We compared DPM and the modified Strube's method using simulated datasets. Simulated datasets were constructed by generating two sets of 10,000 genes with randomly sampled P-values and same directional values (+1). First, we created two sets of input P-values independently of each other (Ind). Uniformly distributed P-values $P_U$ were generated by sampling Z-scores from the normal distribution ($\mu = 0$, $\sigma = 1$) and transforming these to P-values relative to the same normal distribution ($\mu = 0$, $\sigma = 1$). Exponentially distributed P-values $P_E$ were generated by sampling Z-scores from the normal distribution ($\mu = 0$, $\sigma = 1$) and transforming these to P-values relative to ($\mu = 1$, $\sigma = 1$), resulting in an exponential-like distribution that was over-represented in significant P-values (i.e., ~ 25% at P < 0.05). Second, we generated the two sets of input P-values such that the P-values were positively correlated with each other (Cor), by first creating one set of Z-scores as described above (i.e., representing either $P_U$ or $P_E$) and then adding normally distributed noise ($\mu = 0$, $\sigma = 0.2$) to these Z-scores prior to P-value transformation to obtain the second, correlated set of P-values. Spearman correlations of the two sets of P-values were computed. In total, five simulated datasets of P-values were generated: Ind($P_U$, $P_U$), Ind($P_E$, $P_E$), Cor($P_U$, $P_U$), Cor($P_E$, $P_E$), and Ind($P_U$, $P_E$). We then merged the simulated P-values with directional information in three different configurations: all P-values having directional agreement using the CV [ + 1, +1], all P-values having directional disagreement using the CV [ + 1, −1], and half of P-values having directional disagreement and half having directional agreement using the CV [ + 1, +1]. In the latter case, directional values (+1 or −1) were sampled randomly using the binomial distribution. We performed directional analyses of simulated datasets and counted the numbers of significant merged P-values from DPM and modified Strube's methods at different P-value thresholds (0.2, 0.1, 0.05, 0.01).

## Integrating transcriptomics datasets of *HOXA10-AS* in GBM cells

We analysed the genes and pathways prioritised by directional integration of transcriptomics (RNA-seq) data from *HOXA10-AS* lncRNA knockdown (KD) and overexpression (OE) experiments in GBM cells from our earlier study[27]. We used the CV [KD = −1, OE = +1] to prioritise genes with opposite fold-changes in the two experiments to account for the inverse modulation of *HOXA10-AS*. DPM analysis was compared to the non-directional reference analysis that computed merged P-values using Brown's method. We used gene P-values and FC values for 12,996 protein-coding genes from the original study that were filtered previously to exclude lowly expressed genes. Gene sets of biological processes of Gene Ontology (GO)[2] and molecular pathways of Reactome[3] were downloaded from g:Profiler[5] on March 27, 2023. We limited the analysis to gene sets of 10 to 750 genes. All protein-coding genes were used as statistical background. Significantly enriched pathways were selected based on the default multiple testing correction in ActivePathways (FWER < 0.05). Pathways found in the directional and non-directional analyses were merged and visualised as an enrichment map[26] in Cytoscape (v 3.9.1) using standard protocols[1]. Subnetworks were manually organised as functional themes of related pathways. Significant genes in individual pathways were visualised as dot plots with FC and FDR values. Cancer genes of the COSMIC Cancer Gene Census database[53] (v99) were highlighted.

## Integrating cancer proteogenomics data with patient survival information

We integrated quantitative proteomics (isobaric label quantitation analysis with orbitrap) and transcriptomic (RNA-seq) data of cancer samples with patient survival information obtained from the CPTAC-3[10] and TCGA PanCanAtlas projects[7]. This dataset included 1,140 cancer samples of ten cancer types: pancreatic, ovarian, colorectal, breast, kidney, head & neck, and endometrial cancers, two subtypes of lung cancer, and GBM (Supplementary Data 5). Informed consent was obtained from all human participants as part of previous studies. Ethical review was granted by the University of Toronto Research Ethics Board under protocol no. 37521. The main analysis focused on ovarian cancer (OV). We used the combined dataset assembled by Zhang et al.[38] that included transcriptomics data for 15,424 genes and proteomics data for ~10,000 genes that varied between cancer types. We used previously processed transcriptomics and proteomics data represented as standard deviations from cohort median values[38]. First, we derived directional information from transcript or protein associations with overall survival (OS) based on median dichotomisation of transcript or protein expression. Cox proportional-hazards (PH) regression models H0 and H1 were used separately for transcript and protein levels for each gene and in each cancer type. H0 only included clinical covariates as predictors of OS. H1 used transcript or protein expression level together with common clinical covariates (patient age, patient sex, tumor stage) as predictors of OS. H0 and H1 were compared in an ANOVA analysis using chi-square tests, resulting in P-values and HR values for each gene at the protein and transcript level. Resulting matrices of P-values and unit signs from log-transformed HR values were used in directional integration with DPM. Non-directional analysis was conducted using the Brown's method as reference. To handle missing values in input data, genes that had fewer than 20 patients with transcriptomic, proteomic, or clinical information were not analysed and were assigned insignificant values in the input matrices (P-value = 1, direction = 0). The CV [RNA = +1, protein = +1] prioritised genes with matched OS associations at transcript and protein level and penalised genes with opposite OS associations. Pathway enrichment analysis was performed similarly to the *HOXA10-AS* dataset described above. The background set for pathway analysis included 9064 genes for which both transcriptomic and proteomic measurements were available. Significant pathways were selected using the more sensitive FDR correction (FDR < 0.05) instead of the default FWER correction to account for reduced statistical power of OS associations in heterogeneous clinical datasets.

## Integrating RNA-seq, proteomics, and DNA methylation in GBM

We integrated three data modalities with multi-directional constraints: transcriptomics (RNA-seq), quantitative proteomics (isobaric label quantitation analysis with orbitrap), and DNA methylation (CpG Illumina 450k microarray). Transcriptomics and DNA methylation datasets were retrieved from TCGA[7] and proteomics data from CPTAC-3[10]. GBMs with *IDH1* R132H mutations were identified from the Genomic Data Commons (GDC) web portal using TCGA patient IDs. First, we performed differential analyses of transcriptomics, methylation, and proteomics datasets by comparing subsets of GBMs based on *IDH1* mutation status. We limited the

the analyses to 10,902 genes for which all three data types were available. Transcriptomics data were downloaded as gene read counts of transcripts per million (TPM) values using the TCGAbiolinks R package[54] (May 9th, 2023). We compared the transcriptomes of 7 *IDH1*-mutant (*IDH1* R132H) GBMs and 166 *IDH1*-wildtype GBMs. One GBM sample with a different *IDH1* mutation (R132G) was excluded from all analyses. Differential gene expression analysis of *IDH1*-mutant *vs.* *IDH1*-wildtype GBMs was performed non-parametrically using Mann-Whitney U-tests. The resulting P-values were corrected for multiple testing using the Benjamini-Hochberg FDR method. DNA methylation data were downloaded using TCGAbiolinks[54] for six *IDH1*-mutant GBMs and 149 *IDH1*-wildtype GBMs as beta values measuring CpG site methylation. We limited the analysis to CpGs in gene promoters using Human EpicV2 annotations. For each gene, we calculated the mean beta value across the CpG probes in its promoter and conducted a differential methylation analysis of the mean values in *IDH1*-mutant *vs.* *IDH1*-wildtype GBMs using Mann-Whitney U-tests. P-values were corrected for multiple testing using FDR. Genes with significant but small fold-changes in differential methylation (absolute log2FC < 0.25) were soft-filtered by assigning insignificant P-values (P = 1). Proteomics dataset for GBMs was retrieved from the CPTAC-3 project and the dataset processed by Zhang et al. [38]. GBMs carrying *IDH1* R132H mutations were identified in GDC using CPTAC-3 IDs. Significant proteome-wide differences in six *IDH1*-mutant GBMs (*IDH1* R132H) relative to 92 *IDH1*-wildtype GBMs were evaluated using Mann-Whitney U-tests and P-values corrected for multiple testing using FDR. Gene- and pathway-based multi-omics data integration of the *IDH1*-mutant GBM analysis was performed similarly to the analyses above. P-values from transcriptomic, methylation, and proteomic data were merged using DPM as well as the Brown method for reference. Unadjusted P-values and log2-transformed FC values were used for data integration. We prioritised genes with direct associations between transcriptomic and proteomic values and inverse associations with DNA methylation in promoters using the CV [methylation = +1, mRNA = −1, protein = −1]. Pathway enrichment analysis was performed similarly to the analyses described above. The statistical background set for pathway analysis included only the genes detected in all three data types. Significant pathways were selected using ActivePathways at default thresholds (Holm FWER < 0.05). Genes with significant differences in the three datasets were studied using hierarchical clustering and visualised as a heatmap. The heatmap showed unadjusted P-values from the three datasets that were merged non-directionally using Brown's method, corrected for multiple testing using FDR, and filtered for significance using a stringent cut-off (FDR < 0.001). Complete hierarchical clustering was performed using a Euclidean distance metric on directional gene scores (*i.e.*, −log10(FDR) x sign(log2FC)). Using P-value integration from DPM and the non-directional Brown merging, we categorised the selected genes as directionally consistent or inconsistent in the three omics datasets. Known cancer genes from the COSMIC Cancer Gene Census database[53] were labelled.

### Validating pathways found in *IDH*-mutant glioma in additional samples

To validate our pathway enrichment analysis of *IDH1*-mutant gliomas from TCGA and CPTAC, we repeated the analysis in independent glioma samples using transcriptomics (RNA-seq), quantitative proteomics (isobaric label quantitation analysis with orbitrap), and DNA methylation data (CpG Illumina 450k microarray). For transcriptomics and DNA methylation, we compared *IDH1/2*-mutant and *IDH1/2*-wildtype gliomas from the Glioma Longitudinal Analysis (GLASS) cohort[48]. For proteomics data, we compared *IDH1*-mutant and *IDH1*-wildtype gliomas from the study by Oh et al. (2020)[49]. We

limited the analyses to 3,134 genes for which all three data types were available. Transcriptomics, DNA methylation, and patient clinical data from GLASS were downloaded from Synapse (February 24th, 2024). To derive an independent sample set, we excluded TCGA samples from the GLASS dataset according to the project description in the clinical annotations. This resulted in a sample set comprising GBMs (73%), astrocytomas (8%), oligoastrocytomas (5%), oligodendrogliomas (5%), and gliomas of unclassified histology (9%). *IDH* gene mutation status was determined from the idh_codel_subtype column in the clinical table. We compared the transcriptomes of 33 *IDH*-mutant gliomas and 136 *IDH*-wildtype gliomas in a differential gene expression analysis using non-parametric Mann-Whitney U-tests. Gene P-values were corrected for multiple testing using FDR. DNA methylation data from GLASS included 23 *IDH*-mutant GBMs and 99 *IDH*-wildtype GBMs with beta values measuring CpG site methylation. We limited the analysis to CpGs in gene promoters using Human EpicV2 annotations. For each gene, we calculated the mean beta value across the CpG probes in its promoter and conducted a differential methylation analysis of the mean values in *IDH1/2*-mutant vs. *IDH1/2*-wildtype GBMs using Mann-Whitney U-tests. P-values were corrected for multiple testing using FDR. Proteomics data and clinical sample annotations for GBMs in the study by Oh et al.[49] were obtained from the ProteomeXchange portal[55]. *IDH1* mutation status was identified from the "IDH1_mut" field in the clinical annotations. Significantly differentially expressed proteins in 6 *IDH1*-mutant GBMs relative to 48 *IDH1*-wildtype GBMs were evaluated using Mann-Whitney U-tests and P-values were corrected for multiple testing using FDR. Gene- and pathway-based multi-omics data integration was performed similarly to the analyses above. P-values from transcriptomics, methylation, and proteomics data were merged using DPM and using unadjusted P-values and log2-transformed FC values. The Brown method was used as reference. The CV was defined as [methylation = +1, mRNA = −1, protein = −1] similarly to the analysis above. The background set of 3134 genes was used for pathway analysis. Significant pathways were selected in ActivePathways using default thresholds (Holm FWER < 0.05). Gene sets were limited to 10 to 750 genes. This validation analysis combined the three data modalities from two different studies, considered a heterogeneous set of gliomas, included fewer genes and proteins due to limited coverage of proteomics data, and compared results at the level of pathways. These biological and technical aspects of the validation analysis may explain differences we observed.

### Reporting summary

Further information on research design is available in the Nature Portfolio Reporting Summary linked to this article.

## Data availability

Datasets resulting from multi-omics data integration and pathway analyses are provided as Supplementary Data. Most input datasets required to generate and visualise the results are available on GitHub (https://github.com/reimandlab/DPM_publication_code). Input datasets from the GLASS project representing transcriptomics, methylation, and clinical profiles of gliomas are controlled-access and require additional approval by GLASS. GLASS datasets can be retrieved from the Synapse database at https://www.synapse.org (accession numbers: mRNA: syn31121291; clinical: syn31121219; methylation: syn23594913).

## Code availability

The DPM method is available as part of the ActivePathways R package in the CRAN repository (https://cran.r-project.org/web/packages/ActivePathways/) and on GitHub (https://github.com/reimandlab/ActivePathways). Custom scripts to prepare the input data and visualise the results as well as most input datasets are available on GitHub

(https://github.com/reimandlab/DPM_publication_code). Source code of ActivePathways used in this study has been archived in Zenodo[56].

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

## Acknowledgements

We thank Dr. Shraddha Pai and Dr. Michael M. Hoffman for helpful discussions. This work was supported by the Discovery Grant of the Natural Sciences and Engineering Research Council (NSERC) (RGPIN-2023-04646 to J.R.), New Investigator Award of the Terry Fox Research Institute (TFRI) (TFRI-PROJECT-1095 to J.R.), the Canadian Institutes of Health Research (CIHR) Project Grant (PJT-162410 to J.R.), and the Investigator Award to J.R. from the Ontario Institute for Cancer Research (OICR). Funding to OICR is provided by the Government of Ontario. M.S. and M.B. were partially supported by Medical Biophysics fellowships from University of Toronto. A.T.B. was partially supported by the Ontario Graduate Scholarship (OGS). Data used in this publication were partially generated by the Clinical Proteomic Tumor Analysis Consortium (NCI/NIH). The results published here are in part based upon data generated by the TCGA Research Network: https://www.cancer.gov/tcga.

## Author contributions

M.S. developed the method and the software package and performed method benchmarking. M.S. and A.T.B. analysed and interpreted the data. Z.P.K. and M.B. contributed to data analysis and interpretation. L.J.S. contributed to method development and benchmarking. M.S., A.T.B. and J.R. wrote the manuscript. J.R. conceptualised and supervised the project and acquired funding. All authors edited and reviewed the manuscript.

## Competing interests

The authors declare no competing interests.
