## [Peer Review File · Nature Communications]

Directional integration and pathway enrichment analysis for multi-omics dataREVIEWER COMMENTS

Reviewer #1 (Remarks to the Author):

Summary of the paper:

The authors introduce a framework, directionality p-value merging (DPM), enabling a comprehensive joint analysis of diverse omics datasets and patient metadata to identify genes and pathways associated with a specific experimental condition. The novelty of the paper is the integration of discrete directionality information alongside significance estimates in order to prioritise significant genes and pathways with consistent directionality information and inversely penalise genes with conflicting directionality results. Downstream, a ranked pathway enrichment analysis is performed to identify the pathways with the most enriched portion of the gene list and determine which input omics datasets contribute to the enrichment of each pathway. Finally, the enriched pathways are presented visually as an enrichment map.

Using simulated benchmarking data, the authors demonstrate that integrating directional information results in better results than existing p-value merging methods without directional information. Afterwards, the authors conducted three case studies with different omics modalities and patient data input to showcase the versatility of the approach to address different biological and clinical questions such as biomarker identification and over-expression/knockout experiments. Overall the manuscript is well-written and the methods section is presented in a clear and understandable manner. While the benefits and methodological limitations and simplifications are clearly stated, incorporating a specific example of what kind of potentially important findings might be overlooked when utilising DPM would be beneficial. The data and computational code underlying their findings are fully available. Moreover, the authors integrated their method into a well-documented R package with a vignette on how to use DPM.

Major:

None

Minor Comments:

1. The evaluation of the methodology's output is predominantly descriptive. While the absence of ground truth data makes it challenging to conduct a rigorous quantitative validation, a thorough investigation of deprioritized genes and pathways through literature review - similar to the current analysis of the most significant prioritised genes and their known roles in cancer or disease - would enhance the validation routine. For instance, Figures 3C-E show that GO processes related to cell death, cell motility, brain development, and oxygen response were penalised in the non-directional analysis, but the study lacks an interpretation of the previously described relevance of these processes in glioma.
2. In Figure 5F, one of the pathways that has been deprioritized by DPM is the FGFR signaling pathway. Previous studies have found that this pathway is among the most commonly altered molecular pathways in gliomas and FGFR inhibition has become an area of interest for potential therapeutic interventions in the last decade (see e.g. <https://www.ncbi.nlm.nih.gov/pmc/articles/PMC9075824/>, <https://www.ncbi.nlm.nih.gov/pmc/articles/PMC7766440/>, <https://www.nature.com/articles/ng.2611>). This appears to contradict the authors' assertion in their interpretation of Figure 5F, where they suggest that deprioritized biological processes from the non-directional reference analysis are less relevant to glioma biology (L366ff.). Revisiting Figure 5F may result in a more refined and differentiated conclusion regarding the importance of deprioritized pathways in glioma biology.

Text/Figures:

- Line 150: Consider rewriting this sentence more specific by naming the databases
- Line 216: DPM (FDR = 8.2×10^{-4}); the FDR value seems to be rounded down incorrectly despite it being 0.000825228.
- Line 501: The sentence is missing the word 'were'.
Positive CVs are denoted as either '+1' or '1'. Using '+1' consistently in the manuscript would enhance clarity and ensure uniformity
- Figure 3D, 4G & 5F: Consider making "Directional pathway losses" and "Directional Pathway gains" bold. Alternatively, green and red background should be briefly mentioned in the legend.
- Figures 3B, 4B, 5C & S3A: When describing scatter plots, the authors refer to genes with directional disagreements as below the diagonal; which is confusing given that most of the significant genes from DPM shown in blue are also below the diagonal and not on the diagonal.
- Figure 3E & 5G: Specifying in the legends as well that known cancer genes were extracted from the COSMIC Cancer Gene Census database would improve comprehensibility as 'known cancer genes' is ambiguous.
- Figure S3: Adding the numbers of prioritised and penalised genes would improve comprehensibility and make the overall appearance of the scatter plots more consistent.

Reviewer #2 (Remarks to the Author):

Omics technologies have facilitated the generation of datasets from diverse modalities. The development of effective methods for integrating this data is crucial for gaining insights into the biological mechanisms of diseases. In this study, Slobodyanyuk et al. have introduced DPM, a data integration method for the directional integration of results derived from multi-omics data. The paper addresses a significant and common issue where the agreement/disagreement in terms of directionality of effects is ignored during data integration, leading to false positive findings. This study marks a notable advancement in addressing this problem. There is still room for improvement, as some factors limit the generalizability of this method, as outlined below.

1 - The constraints vector (CV) which provides an expected directional relationship of the omics datasets is applied to all entities (i.e., genes) similarly. The complexity of the genome biology suggests that assuming identical effects for all entities may oversimplify the intricate nature of gene regulation. An example is when CV is used to give opposite directions to the methylation compared to the gene and protein expressions. Although generally accurate, there are instances where the methylation of a gene promoter and the expression of the corresponding gene or protein exhibit similar directional effects due to additional post-transcriptional or post-translational regulations.

2 - Another limitation is that only those entities (e.g., genes) for which the corresponding measurements exist across all data modalities are considered. In other words, DPM lacks the capability to handle entities that are absent from certain modalities. Notably, a substantial portion of methylation sites is intergenic, with no overlap with gene promoters. Consequently, the proposed data integration model ignores intergenic methylation sites when combining methylation and gene expression data. Similarly, this limitation exists for the mutations that do not overlap with genes in the whole-genome sequencing data, intergenic open chromatin sites identified by ATAC-seq data and non-coding genetic variants identified by GWAS.

The authors have highlighted various limitations in the Discussion section of the paper. It is crucial for users to acknowledge and consider these limitations when applying DPM in their work.

An additional aspect absent from the existing paper is the absence of result validation. It is imperative to conduct thorough computational validations, potentially utilizing alternative datasets. One example is in the case of glioma, where certain genes and pathways have been reported. It is essential to investigate whether comparable genes and pathways are prioritized when employing an independent

dataset. This validation step is crucial for evaluating the reproducibility of the results derived from DPM.

Minor comments:

Page 5 - Line 135: While P-values inform whether an effect exists or not, the P value do not reveal the size of the effect.

Figure 2: There is a mismatch between the figures and legend in panels B and C. In figures, the labels are shown as "normal" distribution, while in the legend they are defined as "exponential" distribution.

Page 18, Line 367: "to provide" should be used instead of "provide".

Page 19, Line 404: The fact that "the P values from original datasets should be well calibrated" is very important. The authors may want to highlight it earlier in the paper.

Page 21, Line 464: It seems that the nominator of the second term should be the square root of " $k - j$ " instead of " $k - (j+1)$ ".

Reviewer #3 (Remarks to the Author):

The authors propose a computational method DPM for directional integration of genes and pathways across multi-omics datasets. They conduct three case studies to demonstrate their framework. The three case studies exemplify the different use cases for their multiomics pathway enrichment method. First, they show case how to get better pathway enrichments for integrating the RNA differential expression upon KO and over expression of a gene, suggesting their results provide more relevant enrichments. The second case study is about integrating RNA and protein levels in a large cancer cohort where their method was able to find more relevant enrichments that agree with clinical outcomes. In the third case study they integrate methylation, RNA and protein data where they could identify pathways consistent with the IDH-mutant glioma phenotype that they were studying.

Overall, this method provides a very useful tool for the larger genomics community to prioritise relevant biological pathways upon integrating multiple datasets. The most useful feature is likely the directionality assessment, that de-prioritizes genes for which directionality does not agree. My main comment is about the limitations of this application. The authors already discuss some limitations in the discussion, which is the assumption of directionality between modalities. I think an additional limitation should be mentioned about the general uncertainties of pathway analyses, which I think will be useful to discuss openly.

Major comments:

1) It was not clear to me whether the method takes the size of gene sets into account for the p-value calculations. If not, is there a bias towards low p-values for pathways with many genes associated with them? If they do take it into account, would be great to mention explicitly.

2) Limitations: The authors nicely describe some limitations of the method. I would suggest to also add a statement about the limitation of pathway analyses in general, i.e. that pathway analyses can be biased towards the well-studied processes and genes

3) I appreciate the explanation of the method on line 111 with the formula they give. The explanation could be better though, first, o_i is not explained, second. it would be helpful to explain this with

example numbers (i.e. it is not obvious on first glance why for the directional parts, the absolute number is taken. Also it is unclear why everything is multiplied by 2.

REVIEWER COMMENTS

Reviewer #1 (Remarks to the Author):

Summary of the paper:

The authors introduce a framework, directionality p-value merging (DPM), enabling a comprehensive joint analysis of diverse omics datasets and patient metadata to identify genes and pathways associated with a specific experimental condition. The novelty of the paper is the integration of discrete directionality information alongside significance estimates in order to prioritise significant genes and pathways with consistent directionality information and inversely penalise genes with conflicting directionality results. Downstream, a ranked pathway enrichment analysis is performed to identify the pathways with the most enriched portion of the gene list and determine which input omics datasets contribute to the enrichment of each pathway. Finally, the enriched pathways are presented visually as an enrichment map.

Using simulated benchmarking data, the authors demonstrate that integrating directional information results in better results than existing p-value merging methods without directional information. Afterwards, the authors conducted three case studies with different omics modalities and patient data input to showcase the versatility of the approach to address different biological and clinical questions such as biomarker identification and over-expression/knockout experiments. Overall the manuscript is well-written and the methods section is presented in a clear and understandable manner.

We thank the reviewer for the summary and positive comments.

While the benefits and methodological limitations and simplifications are clearly stated, incorporating a specific example of what kind of potentially important findings might be overlooked when utilising DPM would be beneficial.

Thanks for this relevant comment that helps clarify the scope of our method. We extended this discussion in our manuscript.

We already provide one specific example in **Supplementary Figure 3**: we analysed genes and pathways jointly up-regulated or down-regulated in *HOXA10*-AS knockdown (KD) and overexpression (OE) experiments, by defining an alternative constraints vector (CV) as [KD = +1, OE = +1]. This CV achieves an opposite effect to the analysis in **Figure 3** in which prioritised genes and pathways had inverse associations in *HOXA10*-AS KD and OE profiles.

To clarify, DPM does not overlook findings *per se*, since CV is defined by the user. This user-defined CV allows researchers to ask more specific questions of multi-omics data. In DPM, the user chooses their hypothesis by defining the CV. For example, the CV [+1,+1] (or equivalently [-1,-1]) finds direct associations between two input datasets, and DPM identifies significant genes and pathways whose profiles fit that CV. To derive genes and pathways that act opposite to the initial hypothesis, the user defines an alternative CV with an opposite effect, such as the CV [-1,+1] (or equivalently [+1, -1]), and DPM identifies genes and pathways displaying inverse associations in the two input datasets. To quote an often-used remark by Dr. G.E.P. Box: *All models are wrong, but some are useful.*

We added a paragraph to explain this at the end of the first section of Results.

Secondly, we now provide another detailed example on the fibroblast growth factor (FGF) signalling pathway in **Supplementary Figure 5** (Reviewer 1, Minor comment 2; see below).

The data and computational code underlying their findings are fully available. Moreover, the authors integrated their method into a well-documented R package with a vignette on how to use DPM.

Major: None

Thanks for the positive feedback.

Minor Comments:

1. The evaluation of the methodology's output is predominantly descriptive. While the absence of ground truth data makes it challenging to conduct a rigorous quantitative validation, a thorough investigation of deprioritized genes and pathways through literature review - similar to the current analysis of the most significant prioritised genes and their known roles in cancer or disease - would enhance the validation routine.

For instance, Figures 3C-E show that GO processes related to cell death, cell motility, brain development, and oxygen response were penalised in the non-directional analysis, but the study lacks an interpretation of the previously described relevance of these processes in glioma.

We agree that these processes have roles in glioma based on prior literature and we added a few relevant references. However, this analysis is specifically about the oncogenic lncRNA *HOXA10-AS* and the transcriptionally regulated pathways caused by *HOXA10-AS* overexpression (OE) or knockdown (KD) in glioma cells. In this analysis, we assumed that pathways directly downstream *HOXA10-AS* are consistently anti-correlated in the two experiments (KD vs. OE).

Our analysis suggests that the pathways mentioned above are regulated by *HOXA10-AS*, but since the identified pathways are not anticorrelated in the two experiments (KD vs. OE of *HOXA10-AS*), the regulatory effect is likely not directly driven by *HOXA10-AS*. Instead, these inconsistent regulatory effects may reflect a feedback loop or an overall stress response downstream of *HOXA10-AS* modulation in KD and OE experiments. We clarified this better in the text and added **Supplementary Figure 2** to show which genes in these pathways were penalised due to inconsistencies in expression patterns.

2. In Figure 5F, one of the pathways that has been deprioritized by DPM is the FGFR signaling pathway. Previous studies have found that this pathway is among the most commonly altered molecular pathways in gliomas and FGFR inhibition has become an area of interest for potential therapeutic interventions in the last decade (see e.g.

<https://www.ncbi.nlm.nih.gov/pmc/articles/PMC9075824/>,

<https://www.ncbi.nlm.nih.gov/pmc/articles/PMC7766440/>, <https://www.nature.com/articles/ng.2611>).

This appears to contradict the authors' assertion in their interpretation of Figure 5F, where they suggest that deprioritized biological processes from the non-directional reference analysis are less relevant to glioma biology (L366ff.). Revisiting Figure 5F may result in a more refined and differentiated conclusion regarding the importance of deprioritized pathways in glioma biology.

Thanks for pointing out this interesting aspect of our analysis. We did not intend to claim that FGFR signalling is not relevant to glioma biology. We have revised this section and added the new **Supplementary Figure 5** to examine this pathway in detail.

To address the comment, we further analysed two FGF receptor pathways we identified to understand why these enrichments were lost in directional data analysis. The stronger signal came from the pathway with 10 enriched genes (*negative regulation of fibroblast growth factor receptor signalling pathway*, $FWER = 0.0031$). Three genes in the pathway showed inconsistent patterns through RNA, protein, and DNA methylation: *FGF2*, *WNT5A*, *SULF1*. The more general pathway *regulation of fibroblast growth factor receptor signalling pathway*, $FWER = 0.048$) with 18 enriched genes was also found. In this pathway, *NRXN1*, *FRF2*, *WNT5A*, *SULF1* showed inconsistent patterns through mRNA, protein, and DNA methylation levels. Specifically, these penalised genes showed higher DNA promoter methylation (a repressive mark) coupled with mRNA and protein upregulation, which conflicted with the pre-specified directional constraints.

As a potential explanation, here we focused on *IDH*-mutant gliomas. Genetic alterations in FGFR genes and deregulation of the FGFR pathway have been described predominantly in the context of *IDH*-wildtype gliomas or pediatric gliomas, less is known about FGFR regulation in adult *IDH*-mutant gliomas. It is possible that the few genes with directional conflicts in the FGFR pathway are not repressed by DNA promoter methylation in in this glioma subtype.

Text/Figures:

Figures 3B, 4B, 5C & S3A: When describing scatter plots, the authors refer to genes with directional disagreements as below the diagonal; which is confusing given that most of the significant genes from DPM shown in blue are also below the diagonal and not on the diagonal.

Thanks for the comment. In the scatterplots, some genes are indeed below the diagonal, however they are still significantly detected in the directional data integration (shown in blue dots). This is because the directionally inconsistent values from multi-omics datasets are statistically not significant individually, and the resulting penalties are so small that the merged P-values remain significant. This is an expected behaviour of our model: genes are weighted based on their statistical significance as well as their direction.

We updated our figure legends to clarify this point and added a small paragraph in the **Methods** section about using adjusted or unadjusted P-values as DPM input. We also updated the scatterplot in Figure 3B to show unadjusted merged P-values rather than adjusted merged P-values, to make it consistent with other scatterplots in our manuscript.

Line 150: Consider rewriting this sentence more specific by naming the databases.

Line 216: DPM (FDR = 8.2×10^{-4}); the FDR value seems to be rounded down incorrectly despite it being 0.000825228.

Line 501: The sentence is missing the word 'were'.

Positive CVs are denoted as either '+1' or '1'. Using '+1' consistently in the manuscript would enhance clarity and ensure uniformity

Figure 3D, 4G & 5F: Consider making “Directional pathway losses” and “Directional Pathway gains” bold. Alternatively, green and red background should be briefly mentioned in the legend.

Figure 3E & 5G: Specifying in the legends as well that known cancer genes were extracted from the COSMIC Cancer Gene Census database would improve comprehensibility as ‘known cancer genes’ is ambiguous.

Figure S3: Adding the numbers of prioritised and penalised genes would improve comprehensibility and make the overall appearance of the scatter plots more consistent.

Thanks for pointing out these important issues. We fixed these in the revised manuscript.

Reviewer #2 (Remarks to the Author):

Omic technologies have facilitated the generation of datasets from diverse modalities. The development of effective methods for integrating this data is crucial for gaining insights into the biological mechanisms of diseases. In this study, Slobodyanyuk et al. have introduced DPM, a data integration method for the directional integration of results derived from multi-omics data. The paper addresses a significant and common issue where the agreement/disagreement in terms of directionality of effects is ignored during data integration, leading to false positive findings. This study marks a notable advancement in addressing this problem. There is still room for improvement, as some factors limit the generalizability of this method, as outlined below.

Thank you for the summary and positive comments!

1 - The constraints vector (CV) which provides an expected directional relationship of the omics datasets is applied to all entities (i.e., genes) similarly. The complexity of the genome biology suggests that assuming identical effects for all entities may oversimplify the intricate nature of gene regulation. An example is when CV is used to give opposite directions to the methylation compared to the gene and protein expressions. Although generally accurate, there are instances where the methylation of a gene promoter and the expression of the corresponding gene or protein exhibit similar directional effects due to additional post-transcriptional or post-translational regulations.

We fully agree with the reviewer that directional constraints are not likely to represent the ground truth for all genes and pathways because molecular biology is much more complex than a single set of directional constraints. However, our omics datasets are also not able to capture all different forms of gene and protein regulation, such as post-transcriptional or post-translational regulation that the reviewer mentions. We have a paragraph in the Discussion section and we updated it in this revision.

However, our analysis method remains valid given these limitations. Since the constraints vector (CV) is defined by the user, he or she can define CV based on their hypothesis and derive genes and pathways where their multi-omics data significantly agrees with the hypothesis defined in the CV. On the other hand, the user can also investigate the genes and pathways where the hypothesis is not met in the data and the genes and pathways are penalised, because this will reveal potentially interesting counter-examples. This can be done simply by defining an alternative CV with opposite directional constraints (e.g., [+1,+1] vs [+1, -1]). We clarified this better in the in the first section of our Results where our method is described.

We believe that our method provides a useful, more structured way to define specific hypotheses in the researcher's area of interest and interrogate multi-omics datasets at a higher resolution than is currently available in other methods.

2 - Another limitation is that only those entities (e.g., genes) for which the corresponding measurements exist across all data modalities are considered. In other words, DPM lacks the capability to handle entities that are absent from certain modalities. Notably, a substantial portion of methylation sites is intergenic, with no overlap with gene promoters. Consequently, the proposed data integration model ignores intergenic methylation sites when combining methylation and gene expression data. Similarly, this limitation exists for the mutations that do

not overlap with genes in the whole-genome sequencing data, intergenic open chromatin sites identified by ATAC-seq data and non-coding genetic variants identified by GWAS.

This is a great point: indeed, our case studies used DNA methylation sites directly annotated to gene promoters for multi-omics data integration. However, this is not a fundamental limitation of our method. We chose to analyse promoter methylation to present a relatively simple case study that relies fewer assumptions and additional datasets (such as tissue-specific enhancer elements). To address this point, we added a sentence in the Discussion section.

To integrate intergenic methylation sites or other gene-regulatory elements (such as ATAC-seq), researchers often take additional steps, such as annotating enhancer elements of genes or studying elements of the three-dimensional genome such as chromatin loops or topologically associating domains. These elements have additional caveats: these are highly tissue-specific and may be available in low resolution. We have analysed distal non-coding mutations via our previously developed data integration approaches by assigning genes to enhancer elements through 3D chromatin interactions (PMID: 31954095). In principle, our data integration method can be applied directly at the level of CpG sites or regulatory elements, however, for pathway analysis these would still need to be mapped to genes since pathway information is annotated to genes rather than non-coding genomic elements.

The authors have highlighted various limitations in the Discussion section of the paper. It is crucial for users to acknowledge and consider these limitations when applying DPM in their work.

Thank you for the comment, we agree. As stated, appropriate preprocessing of the datasets and defining the relevant hypothesis are two crucial aspects of this analysis. We extended our **Discussion** section in the revision.

An additional aspect absent from the existing paper is the absence of result validation. It is imperative to conduct thorough computational validations, potentially utilizing alternative datasets. One example is in the case of glioma, where certain genes and pathways have been reported. It is essential to investigate whether comparable genes and pathways are prioritized when employing an independent dataset. This validation step is crucial for evaluating the reproducibility of the results derived from DPM.

Thank you, this is an excellent point. We added a validation analysis by combining two independent multi-omics studies of gliomas (RNA, protein, DNA-methylation) to relate our analysis in **Figure 5** in independent clinical samples. We obtained an additional set of transcriptomics and DNA-methylation data of gliomas from the GLASS consortium [PMID: 31748746] and proteomics data from another glioma study by Oh *et al* (2020) [PMID: 32620753]. We used DPM and ActivePathways using similar parameters to the our initial analysis (**Figure 5**), except that we used a updated background gene set that accounted for the more limited coverage of proteomics data available in the validation study.

Encouragingly, we found that the validation analysis captured most major enriched pathway themes and cancer hallmark processes found in the original analysis, such as cell adhesion, cell motility, developmental processes, apoptosis, and others. The validation cohort additionally showed an enrichment of immune system processes, while the discovery cohort in the original analysis showed a few signalling pathways involving growth factors.

There are certain caveats to the analysis that may explain the additional distinct pathways identified in the discovery or the validation analysis: (i) as clinical glioma samples with three types of high-throughput data (mRNA, protein, DNA methylation) are not widely available, we combined samples from different glioma subtypes into a heterogeneous cohort to enable this analysis; (ii) the validation analysis used another proteomics dataset from a third publication (Oh et al. 2020) that profiled another distinct set of gliomas that were not profiled in the RNA and DNA methylation datasets from GLASS; (iii) proteomics data in the validation cohort were more limited and fewer proteins were detected compared to the original analysis (~5000 vs ~3000 proteins), therefore we needed to use a more conservative background list of genes; and (iv) direct comparison of pathway enrichments is challenging due to the redundancy of pathway descriptions in databases; therefore, we compared a narrower set of functional themes that we curated from our enrichment maps. These biological and technical aspects of the validation analysis may explain differences we observed at the pathway level.

We added these results to the manuscript in the final section of Results and in **Figure 5H** (copied above). Thanks again for this great comment. The added analysis lends further confidence to our method.

Minor comments:

Page 5 - Line 135: While P-values inform whether an effect exists or not, the P value do not reveal the size of the effect.

Page 18, Line 367: "to provide" should be used instead of "provide".

Page 21, Line 464: It seems that the nominator of the second term should be the square root of "k - j" instead of "k - (j+1)".

Thanks for these great finds! We updated those parts of the manuscript.

Figure 2: There is a mismatch between the figures and legend in panels B and C. In figures, the labels are shown as "normal" distribution, while in the legend they are defined as "exponential" distribution.

We revised the legend accordingly to note that the P-values were indeed distributed exponentially. The P-values were derived from a normal distribution through a common transformation step (as detailed in **Methods**) which led to this initial misleading description. Thanks for pointing out this inconsistency and improving our manuscript.

Page 19, Line 404: The fact that "the P values from original datasets should be well calibrated" is very important. The authors may want to highlight it earlier in the paper.

Great point. We also added this sentence to the first section of Results where we describe our method.

Reviewer #3 (Remarks to the Author):

The authors propose a computational method DPM for directional integration of genes and pathways across multi-omics datasets. They conduct three case studies to demonstrate their framework. The three case studies exemplify the different use cases for their multiomics pathway enrichment method. First, they show case how to get better pathway enrichments for integrating the RNA differential expression upon KO and over expression of a gene, suggesting their results provide more relevant enrichments. The second case study is about integrating RNA and protein levels in a large cancer cohort where their method was able to find more relevant enrichments that agree with clinical outcomes. In the third case study they integrate methylation, RNA and protein data where they could identify pathways consistent with the *IDH*-mutant glioma phenotype that they were studying.

Overall, this method provides a very useful tool for the larger genomics community to prioritise relevant biological pathways upon integrating multiple datasets. The most useful feature is likely the directionality assessment, that de-prioritizes genes for which directionality does not agree. My main comment is about the limitations of this application. The authors already discuss some limitations in the discussion, which is the assumption of directionality between modalities. I think an additional limitation should be mentioned about the general uncertainties of pathway analyses, which I think will be useful to discuss openly.

Thank you for the summary and positive feedback. We agree that modelling directional constraints is a useful feature for prioritising genes and pathways in multi-omics data. We also agree that pathway analysis has general uncertainties and we have discussed these in our revised manuscript.

Major comments:

1) It was not clear to me whether the method takes the size of gene sets into account for the p-value calculations. If not, is there a bias towards low p-values for pathways with many genes associated with them? If they do take it into account, would be great to mention explicitly.

Great point. We recommend using gene sets of intermediate size for our pathway analysis (min 5, max 1000 by default), however the method certainly takes gene set size into account when computing P-values in the ranked hypergeometric test. This test is well established in the field. We added a brief description of our pathway enrichment analysis in the **Methods** section and refer the reader to our prior publications for further details and best-practices [PMIDs: 30664679, 32024846].

Gene set sizes affect P-value calculations in different ways: in Gene Ontology and Reactome, there are many small gene sets (1-3 genes) that are underpowered and often too specific; these tend to increase multiple testing correction and lead to filtering of results. On the other hand, there are some very large and overly general gene sets with 1000s of genes that add little to data interpretation, however these are statistically highly powered and lead to strong P-values. Therefore, we recommend filtering very small and very large gene sets. This is discussed in our review paper and step-by-step protocol on pathway enrichment analysis [PMID: 30664679].

2) Limitations: The authors nicely describe some limitations of the method. I would suggest to also add a statement about the limitation of pathway analyses in general, i.e. that pathway analyses can be biased towards the well-studied processes and genes

Thanks, we agree. We have added a note in the Discussion.

3) I appreciate the explanation of the method on line 111 with the formula they give. The explanation could be better though, first, o_j is not explained, second. it would be helpful to explain this with example numbers (i.e. it is not obvious on first glance why for the directional parts, the absolute number is taken. Also it is unclear why everything is multiplied by 2.

Thanks for the comment, this helped improve our manuscript. We addressed as follows:

(i) We refer to **Supplementary Figure 1** earlier in the manuscript. This shows a reproducible example of directional data integration with actual numbers. This was previously included in Supplementary Figure 2.

(ii) We now explain the formula in more detail, specifically in the **Results** section early in the manuscript. It is also explained in the **Methods** section similarly to our original manuscript.

(iii) We explain multiplication of the values by 2. To develop DPM, we extended Fisher's and Brown's P-value merging techniques that both use the chi-square distribution to evaluate significance and we have adapted the multiplication by two from those methods.

REVIEWERS' COMMENTS

Reviewer #1 (Remarks to the Author):

Great job with the revision and improvements! The limitations of the method are more precisely highlighted, potentially unclear paragraphs have been revised and the added validation study adds robustness to this study. The authors have addressed all the concerns in my review in their revised manuscript.

Reviewer #1 (Remarks on code availability):

The authors provide installation instructions as well as vignettes/examples on how to use their ActivePathways R package. The package is properly versioned with version release information, published under a GPL-3 license and a README file is provided. I was able to install the package and run examples, though I have not tried to reproduce any of the figures.

In their code and data availability section, the authors state that "Custom scripts to prepare and visualise the data [as well as Intermediate files] in bulk form are available from the corresponding author upon reasonable request.". While it seems unnecessary and unfeasible to make the intermediate files accessible, it would be beneficial to have the custom scripts to prepare and visualise the data publicly available, either in a separate GitHub repository, a zip folder or as a Zenodo repository.

Reviewer #2 (Remarks to the Author):

The study by Slobodyanyuk et al. marks a significant advancement in addressing a critical problem in the field. Typically, the directionality of effects is overlooked during data integration, leading to false positive findings. This study successfully addresses this problem. The authors have addressed all my previous comments. In my opinion, this study can be accepted for the publication.

Reviewer #3 (Remarks to the Author):

The authors have addressed my comments very nicely and I congratulate (and thank) them for developing a tool that is likely going to be very useful for the community.

Judith Zaugg

REVIEWERS' COMMENTS

Reviewer #1 (Remarks to the Author):

Great job with the revision and improvements! The limitations of the method are more precisely highlighted, potentially unclear paragraphs have been revised and the added validation study adds robustness to this study. The authors have addressed all the concerns in my review in their revised manuscript.

Reviewer #1 (Remarks on code availability):

The authors provide installation instructions as well as vignettes/examples on how to use their ActivePathways R package. The package is properly versioned with version release information, published under a GPL-3 license and a README file is provided. I was able to install the package and run examples, though I have not tried to reproduce any of the figures. In their code and data availability section, the authors state that "Custom scripts to prepare and visualise the data [as well as Intermediate files] in bulk form are available from the corresponding author upon reasonable request.". While it seems unnecessary and unfeasible to make the intermediate files accessible, it would be beneficial to have the custom scripts to prepare and visualise the data publicly available, either in a separate GitHub repository, a zip folder or as a Zenodo repository.

Reviewer #2 (Remarks to the Author):

The study by Slobodyanyuk et al. marks a significant advancement in addressing a critical problem in the field. Typically, the directionality of effects is overlooked during data integration, leading to false positive findings. This study successfully addresses this problem. The authors have addressed all my previous comments. In my opinion, this study can be accepted for the publication.

Reviewer #3 (Remarks to the Author):

The authors have addressed my comments very nicely and I congratulate (and thank) them for developing a tool that is likely going to be very useful for the community.
Judith Zaugg

We would like to thank all three reviewers for their positive comments and for approving our manuscript for publication.

Regarding Reviewer #1's remarks on code availability:

We now provide a GitHub repository with our source code to process and visualise the data. We also provide a Zenodo archive of the ActivePathways package that contains the DPM method used in the manuscript. We agree that providing intermediate files is not feasible at this stage, especially as these contain traces of some of the restricted-access datasets we use (i.e. GLASS consortium datasets). We sincerely hope our code repository resolve this matter. Thank you again for the positive reviews and constructive comments.